# Variational Task Vector Composition

**Boyuan Zhang**[1]    **Yingjun Du**[2]    **Xiantong Zhen**[3]    **Ling Shao**[1*]

[1]UCAS-Terminus AI Lab, School of Engineering Science,
University of Chinese Academy of Sciences
[2]VIS Lab, University of Amsterdam
[3]Central Research Institute, United Imaging Healthcare, Co., Ltd.
`zhangboyuan23@mails.ucas.ac.cn`

## Abstract

Task vectors capture how a model changes during fine-tuning by recording the difference between pre-trained and task-specific weights. The composition of task vectors, a key operator in task arithmetic, enables models to integrate knowledge from multiple tasks without incurring significant additional inference costs. In this paper, we propose variational task vector composition (VTVC), where composition coefficients are taken as latent variables and estimated in a Bayesian inference framework. Unlike previous methods that operate at the task level, our framework focuses on sample-specific composition. Motivated by the observation of structural redundancy in task vectors, we introduce a Spike-and-Slab prior that promotes sparsity and aims to preserve the most informative components. To further address the high variance and sampling inefficiency in sparse, high-dimensional spaces, we develop a gated sampling mechanism that constructs a controllable posterior by filtering the composition coefficients based on both uncertainty and importance. This yields a more stable and interpretable variational framework by deterministically selecting reliable task components, reducing sampling variance while improving transparency and generalization. Experimental results demonstrate that our method achieves state-of-the-art average performance across a diverse range of benchmarks, including image classification and natural language understanding. These findings highlight the practical value of our approach, offering a new, efficient, and effective framework for task vector composition.

## 1    Introduction

Task vectors represent the difference between a model's pre-trained and fine-tuned weights, capturing the changes during fine-tuning on specific tasks. Using task vectors, task arithmetic [17] enables simple and efficient model editing. The composition of task vectors, a key operator in task arithmetic, enables models to integrate knowledge from multiple tasks without increasing additional inference costs. Task vector composition has demonstrated strong performance across a variety of domains, including computer vision [14], natural language processing [41, 23, 13], and multimodal learning [16, 26]. This approach provides a practical solution for knowledge integration and model editing [48, 44, 53, 46].

In recent years, the composition of task vectors has been widely studied to improve efficiency and controllability [52]. Some works focus on improving task vector representations by redesigning spaces and developing parameterization techniques to better capture the relationships between tasks [30, 9]. Others extend the functional scope of task vectors to reduce computational overhead [41, 14, 15]. Advances in composition methods also enable more effective integration of knowledge across multiple

---

*Corresponding Author (`ling.shao@ieee.org`)

tasks, enhancing both the controllability and interpretability of models [51, 54]. Meanwhile, in the broader field of meta-learning, researchers have also explored using probabilistic frameworks such as variational inference to learn task-adaptive modules, thereby improving the model's generalization ability on new tasks[55, 8]. While recent works have made notable progress, three key challenges remain under-explored. (1) Existing approaches typically rely on deterministic composition schemes, lacking mechanisms to quantify uncertainty. (2) Most methods operate at the task level, which limits their ability to adapt to sample-level variability. (3) Task vector spaces often exhibit redundant structure, which may reduce efficiency in both inference and storage.

In this paper, we propose a novel approach for the **V**ariational **T**ask **V**ectors **C**omposition (**VTVC**). Our main contributions are as follows: *(i)* We cast task vector composition as a variational inference problem and introduce an amortized inference network to model sample-specific posteriors over composition coefficients. This formulation enables efficient integration of domain-specific knowledge at the sample level and overcomes the limitations of conventional task-level composition. *(ii)* To address the structural redundancy in task vector spaces, we introduce a Spike-and-Slab prior that promotes sparsity by modeling the variational posterior as a mixture of zero-valued *spikes* and Gaussian *slabs*. This structured prior guides the variational posterior to focus on essential task components, enhancing interpretability and improving the efficiency of sample-specific composition. *(iii)* To address the high variance and sampling inefficiency of traditional sampling-based inference, we develop a controllable posterior via gated sampling, which deterministically filters composition coefficients based on their estimated uncertainty and importance. By deterministically selecting task coefficients based on uncertainty and importance, the proposed framework avoids the high variance of stochastic sampling, highlights the components that contribute to adaptation, and promotes consistent inference, making it more stable, interpretable, and reliable.

We evaluate our framework on diverse benchmarks in computer vision and natural language processing. Experimental results show that VTVC achieves strong performance in various task arithmetic scenarios, including task addition and task negation. Our analyses of the three main contributions–variational composition, Spike-and-Slab priors, and the gated sampling process highlight the method's advantages. We demonstrate its ability to integrate sample-level knowledge, reduce redundancy, and improve inference stability.

## 2 Related work

**Task vector composition and task arithmetic.** Task arithmetic is a central technique in the model merging field, which aims to combine multiple specialized models into one [17]. The field has been widely studied to improve efficiency and controllability [52], with theoretical foundations established from perspectives like the Neural Tangent Kernel and generalization analysis [30, 23]. Early frameworks like aTLAS introduced anisotropic scaling for more precise control [51]. A key challenge is managing interference between task parameters. Recent work has largely focused on two directions. The first is leveraging sparsity, where methods like DARE merge only the critical, sparse parameters to improve efficiency [50, 43, 11]. The second direction focuses on directly resolving task interference, with methods aligning parameter signs [47] or redesigning task vector spaces to find less conflicting representations [9, 54, 4]. The functional scope of task vectors has also been extended in various applications, from vision to language models [14, 41, 15]. Other approaches achieve sample-level adaptation by modifying the model architecture uses Mixture-of-Experts (MoE) modules to dynamically route inputs to different task vectors [38]. While these methods are effective, they are largely deterministic. In contrast, we propose a probabilistic inference framework that learns sample-specific composition coefficients, uniquely allowing it to quantify and leverage uncertainty without requiring architectural changes.

**Applications of Spike-and-Slab priors.** Spike-and-Slab priors are widely used in Bayesian sparse modeling to distinguish between important and unimportant variables [21, 2]. The foundational work by Mitchell et al. [28] introduced them as mixture distributions for effective variable selection. While traditional estimation relied on MCMC sampling [18], the need for scalability on large datasets has prompted a shift toward more efficient variational inference (VI) approaches [40, 34]. In deep learning, these priors have been applied to tasks like unsupervised image classification [10], with recent studies also exploring structured forms to model complex correlations [37, 1, 36, 27]. However, their practical integration with large deep models remains a challenge, especially for sample-specific sparse inference.

## 3 Preliminary

In transfer learning and multi-task learning, task-specific knowledge is often captured by modeling changes in network parameters, including models based on CLIP [31], GPT-2 [32], and T5 [33]. In this paper, we focus particularly on CLIP, as it offers powerful multimodal representations and aligns with experimental settings used in previous work.

Formally, denote the CLIP image encoder as a function $f : \mathcal{X} \times \Theta \to \mathcal{Z}$, where for an input image $x \in \mathcal{X}$ and parameters $\theta \in \Theta$, the output $z = f(x; \theta)$ represents the learned latent representation for the input image. Given the pre-trained parameters $\Theta_0$ and the fine-tuned parameters $\Theta_t$ in a downstream task $t$, the corresponding task vector is defined as $\tau_t = \Theta_t - \Theta_0$. In practice, we follow standard procedures by fine-tuning only the image encoder while keeping the text encoder fixed, which helps maintain feature alignment across tasks.

Task vectors can be composed through various arithmetic operations, and the most common operation is task addition. In task addition, given a task vector pool $\mathcal{T}$ with $N$ task vectors $\{\tau_i\}_{i=1}^N$ fine-tuned on different tasks, with a global learnable scaling factor $\lambda$, a unified model can be constructed as: $\Theta_{\text{new}} = \Theta_0 + \lambda \cdot \sum_{i=1}^N \tau_i$. This form of composition has shown strong practical value, enabling effective integration of knowledge from multiple tasks into a single model and demonstrating superior transfer capabilities in image classification and transfer learning benchmarks.

Task vectors can also be decomposed into block-wise representations. Based on functional partitioning, aTLAS [51] divides each task vector $\tau_i$ into $M$ parameter blocks $\{\tau_i^j\}_{j=1}^M$, and assigns an independent scaling coefficient $\Lambda_i^j$ to each block. Such modular decomposition also enables the exploration of task relevance at different network layers or functional units, which can be further exploited in downstream analysis. The composed model parameters can then be written as:

$$\Theta = \Theta_0 + \sum_{i=1}^N \sum_{j=1}^M \Lambda_i^j \cdot \tau_i^j. \tag{1}$$

This block-wise formulation enables finer-grained knowledge integration and provides a foundation for efficient multi-task learning and task transfer. While most existing methods perform task vector composition at the task level with fixed coefficients, they fail to capture fine-grained differences between samples. To address this limitation, we model composition at the sample level, where the coefficients become functions of the input $x$. Formally, the parameters of a sample-specific merged model $\Theta(x)$ can be written as:

$$\Theta(x) = \Theta_0 + \sum_{i=1}^N z_i(x) \cdot \tau_i \quad \text{(non-block case)}, \tag{2}$$

$$\Theta(x) = \Theta_0 + \sum_{i=1}^N \sum_{j=1}^M z_{ij}(x) \cdot \tau_i^j \quad \text{(block case)}. \tag{3}$$

Here, $z_i(x)$ and $z_{ij}(x)$ are sample-specific coefficients produced by an inference network that takes $x$ as input. This finer-grained approach allows the model to adapt to input variability and enhances performance in heterogeneous environments, underpinning our proposed variational framework.

## 4 Methodology

This section introduces our proposed framework for the variational composition of task vectors. In Section 4.1, we reformulate the composition problem as a variational inference process and introduce an amortized inference network to model sample-specific posteriors over composition coefficients. To address the structural redundancy in task vector spaces, we then introduce a Spike-and-Slab prior to promote sparsity by modeling the variational posterior as a mixture distribution in section 4.2. We also propose a controllable posterior with gated sampling that automatically selects informative task vectors in section 4.3, which enhances the model's generalization capability and robustness.

## 4.1 Variational Composition of Task Vectors

Conventional task vector composition is typically performed as a direct and deterministic modification of model weights, lacking any mechanism for uncertainty quantification. As a result, such approaches are less capable of adapting to sample-specific variations and cannot estimate uncertainty in the composition process. In our variational inference framework, the composition coefficients $\mathbf{z}$ are treated as latent variables, while the input $\mathbf{x}$ and labels $\mathbf{y}$ are observed. Figure 1 illustrates the probabilistic dependencies. According to Bayesian theory, our objective is to compute the posterior distribution $p(\mathbf{z}|\mathbf{x}, \mathbf{y})$. Since the posterior is typically difficult to compute directly, variational inference introduces a parameterizable approximate posterior distribution $q(\mathbf{z}|\mathbf{x})$, leading to the evidence lower bound (ELBO):

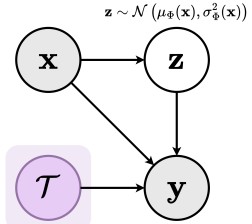

$$\log p(\mathbf{y}|\mathbf{x}) \geq \mathbb{E}_{q(\mathbf{z}|\mathbf{x})}[\log p(\mathbf{y}|\mathbf{x}, \mathbf{z})] - D_{\mathbf{KL}}(q(\mathbf{z}|\mathbf{x})||p(\mathbf{z})). \quad (4)$$

Here, the first term represents the expected log-likelihood, which promotes the model to make accurate predictions, while the second term is the KL divergence, which regularizes the posterior towards

**Figure 1: Variational Composition of Task Vectors.**

the prior to prevent overfitting. This framework naturally recasts the composition problem as an inference task, allowing us to incorporate flexible prior knowledge, such as sparsity or structural constraints, into the composition process. The detailed mathematical derivation is provided in Appendix A.1.

To efficiently realize sample-level variational inference, we employ amortized inference by designing a neural network with parameters $\Phi$ to parameterize the posterior distribution: $q_\Phi(\mathbf{z}|\mathbf{x}) = \mathcal{N}(\mathbf{z}; \mu_\Phi(\mathbf{x}), \sigma_\Phi(\mathbf{x}))$. This network takes samples $\mathbf{x}$ as inputs and predicts the parameters of the composition coefficient distribution $\mathbf{z}$. Amortized inference improves both efficiency and scalability by sharing parameters across samples.

In summary, sample-level composition allows the model to adapt composition coefficients to individual inputs, offering finer control over task integration and leading to improved generalization in heterogeneous settings. This represents a fundamental difference from task-level approaches. However, conventional variational inference typically uses Gaussian priors, which are unable to capture or address structural redundancy in task vector spaces. This can lead to inefficient composition and increased risk of overfitting.

## 4.2 Spike-and-Slab Priors for Sparse Representations

The standard Gaussian prior is widely used, but it fails to capture the structural properties inherent in high-dimensional task vector spaces. As illustrated in Fig. 2, we conduct a thorough redundancy analysis of task vector representations. Using t-SNE and singular value decomposition, we find significant redundancy in task vectors, which leads to inefficient integration and a higher risk of overfitting. For instance, over 95% of the variance can be explained by fewer than 40 principal components across all datasets, highlighting substantial compressibility. Moreover, when input samples are located near the boundaries of multiple task domains, Gaussian priors are unable to capture the multimodal nature of the underlying composition, thereby limiting model performance on heterogeneous data.

Our variational composition framework offers a flexible way of incorporating prior knowledge to address the issue of structural redundancy in task vector representations. We introduce a Spike-and-Slab prior, which effectively promotes structural sparsity in the compositions of task vectors. The Spike-and-Slab prior is defined as a mixture distribution:

$$p(\mathbf{z}) = (1 - \pi)\,\delta_0(\mathbf{z}) + \pi\,\mathcal{N}(\mathbf{z}; 0, \sigma^2), \quad (5)$$

where $\delta_0(\mathbf{z})$ denotes the Dirac delta distribution at zero, $\mathcal{N}(\mathbf{z}; 0, \sigma^2)$ denotes a Gaussian distribution with zero mean and variance $\sigma^2$, and $\pi$ is the mixture coefficient that controls the expected sparsity of the coefficients. The Spike-and-Slab prior allocates probability mass between a point mass at zero (the *spike*) and a continuous Gaussian distribution (the *slab*). This ensures that coefficients are either exactly zero or drawn from the Gaussian, thereby enabling explicit and interpretable structure selection. By promoting coefficient sparsity, the Spike-and-Slab prior automatically identifies and retains the most important task vector components while effectively eliminating irrelevant ones. This

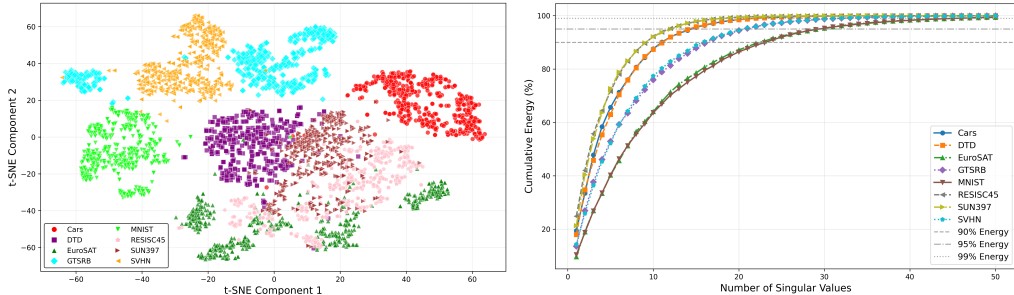

(a) t-SNE clustering of task representations.  (b) Cumulative singular value energy distribution.

**Figure 2: Redundancy and structural analysis of task vectors.** (a) t-SNE visualization shows that task representations form distinct clusters corresponding to each dataset, but there are notable overlap regions, indicating partial coupling of task features across domains. (b) Cumulative singular value energy plots reveal that fewer than 40 principal components account for over 95% of the variance across all eight datasets. These analyses reveal substantial structural redundancy in task vector representations.

property not only improves model efficiency but also enhances interpretability, allowing us to clearly determine which task knowledge is relevant to each input sample.

Under the Spike-and-Slab prior, we adopt a factorized variational posterior:

$$q_\Phi(\mathbf{z}|\mathbf{x}) = \prod_i \prod_j q_\Phi(\omega_i^j|\mathbf{x}) q_\Phi(\mathbf{z}_i^j|\omega_i^j, \mathbf{x}), \tag{6}$$

Specifically, for each composition coefficient $\mathbf{z}_i^j$, we introduce a corresponding binary indicator variable $\omega_i^j$. The coefficient is retained when $\omega_i^j = 1$ and set to zero when $\omega_i^j = 0$.

By performing variational inference, we obtain the following ELBO objective:

$$\mathcal{L}_{\textbf{ELBO}} = \mathbb{E}_{q_\Phi(\mathbf{z},\omega|\mathbf{x})}[\log p(\mathbf{y}|\mathbf{x}, \mathbf{z}, \omega)] - D_{\textbf{KL}}(q_\Phi(\mathbf{z}, \omega|\mathbf{x}) || p(\mathbf{z}, \omega)), \tag{7}$$

To optimize this framework, we employ amortized inference with Monte Carlo sampling. We design a neural network parameterized by $\Phi$ that inputs sample $\mathbf{x}$ and outputs three sets of parameters: inclusion probabilities $\pi_i^j(\mathbf{x})$, weight means $\mu_{ij}(\mathbf{x})$, and log variances $\log \sigma_{ij}^2(\mathbf{x})$. During training, we first sample binary inclusion variables $\omega_i^j \sim \text{Bernoulli}(\pi_i^j(\mathbf{x}))$, then sample weights $\mathbf{z}_i^j \sim \mathcal{N}(\mu_{ij}(\mathbf{x}), \sigma_{ij}^2(\mathbf{x}))$, and finally combine them to obtain sparse composition coefficients $\tilde{\mathbf{z}}_i^j = \omega_i^j \cdot \mathbf{z}_i^j$. The complete derivation of the variational objective with Spike-and-Slab prior is available in Appendix A.2.

However, in high-dimensional settings, Monte Carlo-based variational inference often suffers from unstable sparsity patterns and unreliable results [35, 6]. This limitation motivates the development of a more stable and controllable posterior, which we describe in the following section.

### 4.3 Controllable Posterior with Gated Sampling

To address the instability and high variance of Monte Carlo sampling in high dimensions, we propose a gated sampling mechanism for controllable posterior. Instead of sampling binary variables from the conventional Bernoulli distribution with probability $\pi(\mathbf{x})$, our method employs a deterministic, continuous gating function to modulate composition coefficients. The continuous gating function replaces discrete sampling with differentiable alternative, enabling gradient-based training while preserving the structure of variational inference. Figure 3 illustrates both stochastic and deterministic inference processes, highlighting how our method transitions from random Bernoulli sampling to uncertainty-aware, soft-gated posterior construction.

We begin by constructing an uncertainty estimate $\mathcal{U}$, which integrates two components: gradient sensitivity and distributional deviation. Gradient sensitivity $\mathcal{S}$ quantifies how responsive a coefficient is to small perturbations in the input, while distributional deviation $\mathcal{V}$ measures the extent to which a coefficient deviates from the batch mean, where $\mu_B$ and $\sigma_B$ denote the mean and standard deviation

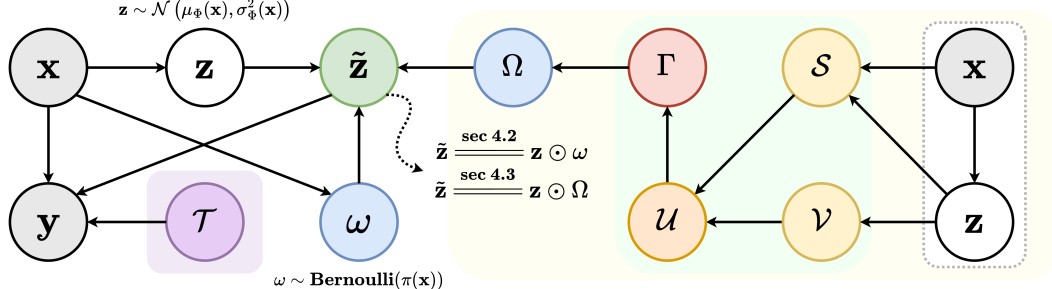

**Figure 3: Variational Composition of Task Vectors with Spike-and-Slab Prior**. The framework illustrates two inference approaches based on the Spike-and-Slab prior. On the left, the sampling-based variational posterior process generates latent variables $\mathbf{z}$ from input features $\mathbf{x}$ via $\mathcal{N}(\mu_\Phi(\mathbf{x}), \sigma_\Phi^2(\mathbf{x}))$, producing sparse representations $\tilde{\mathbf{z}} = \mathbf{z} \odot \omega$ through element-wise multiplication with binary indicators $\omega$ sampled from Bernoulli$(\pi(\mathbf{x}))$. On the right, the deterministic gated sampling process computes uncertainty $\mathcal{U}$ from gradient sensitivity $\mathcal{S}$ and distributional deviation $\mathcal{V}$, uses an adaptive threshold $\Gamma$ to generate continuous gating variables $\Omega$, and produces sparse representations $\tilde{\mathbf{z}} = \mathbf{z} \odot \Omega$. This framework demonstrates the transition from random binary selection to deterministic continuous gated in the inference process.

of the coefficient within the batch, respectively. The overall uncertainty measure is formulated as a weighted combination:

$$\mathcal{U} = \eta \cdot \mathcal{S} + (1 - \eta) \cdot \mathcal{V} = \eta \cdot |\nabla_\mathbf{x}\mathbf{z}(\mathbf{x})|_2 + (1 - \eta) \cdot \left| \frac{\mathbf{z}(\mathbf{x}) - \mu_B}{\sigma_B} \right|, \tag{8}$$

where $\eta$ is a balancing coefficient. Intuitively, coefficients that are highly sensitive to input changes or show large fluctuations among similar samples are deemed unreliable and should be suppressed, while stable and significant coefficients should be preserved.

We then define an adaptive gating function:

$$\mathcal{G}(\mathbf{z}, \mathcal{U}; \Psi) = \mathbf{z} \cdot \mathbb{I}(|\mathbf{z}| \geq \Gamma(\mathcal{U}; \Psi)), \tag{9}$$

where the threshold function $\Gamma(\mathcal{U}; \Psi)$ varies with the uncertainty metric $\mathcal{U}$, creating an adaptive selection mechanism as follows:

$$\Gamma(\mathcal{U}; \Psi) = \psi_1 \cdot (1 + \psi_2 \cdot \mathcal{U}), \tag{10}$$

where $\Psi = \{\psi_1, \psi_2\}$ are learnable parameters, $\psi_1$ is a base threshold, and $\psi_2$ controls sensitivity to uncertainty. Under this mechanism, coefficients with high uncertainty are subjected to higher thresholds and thus are less likely to be selected, while those with low uncertainty face lower thresholds, increasing their likelihood of being retained. This adaptive selection ensures that more reliable and informative components are preferentially integrated into the final model composition.

To enable end-to-end differentiable training, we approximate the hard-gated function with a soft-gated function, thereby replacing the Monte Carlo sampling methods described in Section 4.2 with a deterministic point estimation approach:

$$\Omega(\mathbf{z}, \mathcal{U}) = \sigma \left( \frac{|\mathbf{z}| - \Gamma(\mathcal{U}; \Psi)}{\rho} \right), \tag{11}$$

where $\sigma(\cdot)$ denotes the sigmoid function and $\rho$ is a temperature hyperparameter. During training, the soft-gated function allows for gradient-based optimization, while at inference time we switch to hard thresholding to ensure sparse and interpretable coefficient selection. Based on this design, each coefficient's variational posterior can be approximated as a mixture of Spike-and-Slab priors:

$$q(\mathbf{z}|\mathbf{x}) = [1 - \Omega(\mathbf{z}, \mathcal{U})] \cdot \delta_0(\mathbf{z}) + \Omega(\mathbf{z}, \mathcal{U}) \cdot \mathcal{N}(\mathbf{z}; \mu_\Phi(\mathbf{x}), \sigma_\Phi^2(\mathbf{x})). \tag{12}$$

Here, $\Omega$ replaces the binary indicator variable $\omega$ for deterministic gating. To further address the high variance of sampling in high-dimensional spaces, we shift from sampling the coefficients $\mathbf{z}$ to predicting them as a deterministic point estimate from the input $\mathbf{z} = f_\Phi(\mathbf{x})$. In our framework, instead of modeling a distribution over $\mathbf{z}$, we explicitly compute an uncertainty metric $\mathcal{U}$ and use it to control the gating function $\Omega$. This leads to a more stable training process. We formalize this deterministic choice for $\mathbf{z}$ within the variational posterior using a Dirac delta distribution as follows:

$$q_{\Phi, \Psi}(\Omega, \mathbf{z}|\mathbf{x}) = q_\Psi(\Omega|\mathbf{z})\delta_{\mathbf{z}=f_\Phi(\mathbf{x})}. \tag{13}$$

This parameterization enables the selection of deterministic coefficients within the variational inference framework, while explicitly modeling uncertainty.

In addition to the primary objective of the ELBO, we introduce auxiliary regularization losses $\mathcal{L}_{\text{reg}}$ to further stabilize training and encourage exploration. These include a boundary loss, penalizing coefficients near the threshold to encourage more decisive selection decisions, and an exploration loss, promoting threshold-related parameters to search in a broader space to avoid local optima. Detailed definitions of these regularization terms and hyperparameter settings are provided in Appendix B.

In summary, the final training objective can be expressed as:

$$\underset{\Phi, \Psi}{\arg\min} \frac{1}{|\mathcal{D}_t|} \sum_{s=1}^{|\mathcal{D}_t|} \mathcal{L}\left(f\left(\mathbf{x_s}; \Theta_0 + \mathcal{G}\left(\mathbf{z}(\mathbf{x_s}; \Phi), \mathcal{U}; \Psi\right) \mathcal{T}\right), \mathbf{y_s}\right) + \lambda \mathcal{L}_{\text{reg}}(\mathbf{z}, \mathcal{U}; \Psi), \tag{14}$$

where $\lambda$ is the balance parameter. Our final training objective comprises two components: the primary prediction loss based on the gated sampling, and a regularization term that stabilizes the optimization process while enhancing the model's generalization capacity. By modeling coefficient uncertainty explicitly, our method enhances generalization and interpretability. Overall, this uncertainty-guided, gated composition yields a more robust and interpretable mechanism for knowledge integration across diverse tasks.

# 5 Experiments

## 5.1 Experiments Setup

**Tasks and Datasets.** We evaluate our framework across three distinct scenarios. For multi-task model merging, we follow previous work and use eight image classification datasets: Cars [20], DTD [5], EuroSAT [12], GTSRB [39], MNIST [22], RESISC45 [3], SUN397 [45], and SVHN [29]. For NLP applications, we use the General Language Understanding Evaluation(GLUE) benchmark [42]. We also design a task negation experiment to evaluate performance on more complex task arithmetic.

**Baselines.** We compare our method VTVC against a comprehensive set of recent model merging methods, including Task Arithmetic [17], aTLAS [51], TIES-Merging [47], DARE [50], AdaMerging [49], and WUDI-Merging [4]. To ensure a fair comparison, all methods are evaluated under the same experimental conditions.

**Implementation Details.** We use three pre-trained CLIP Vision Transformer (ViT) architectures: ViT-B/16, ViT-B/32, and ViT-L/14 [31, 7]. For NLP tasks, we use RoBERTa [24] as the backbone model. All experiments are trained with the AdamW optimizer [25]. For our method's gated posterior, we set the base threshold($\psi_1$) to 0.05 and the sensitivity parameter($\psi_2$) to 1.0. All experiments were conducted on eight NVIDIA A40 GPUs. Further details on hyperparameters and task-specific settings are provided in Appendix B.

**Evaluation Metrics.** For model merging and image classification tasks, we report the top-1 accuracy. For task negation, we evaluate the accuracy on both the forgotten target task and the remaining control tasks. For the GLUE benchmark, we report the average score across all tasks. We also report the gated ratio of our method, which measures the average proportion of active coefficients, to analyze its sparsity.

## 5.2 Main Results

**Performance on Multi-Task Merging.** We first evaluated VTVC against current state-of-the-art methods on model merging tasks. The evaluation covered eight visual tasks using ViT-B/32 models and eight NLP tasks using RoBERTa-base models. All methods were tested under fair experimental settings. As shown in Table 1 and Table 2, VTVC achieved the highest average accuracy in both domains. For visual tasks, VTVC reached 87.45% average accuracy, with particularly strong performance on DTD and RESISC45 datasets, improving accuracy by 15.35% and 5.14% respectively compared to the previous best method. For NLP tasks, VTVC achieved 81.93% average accuracy, showing significant improvements on challenging tasks such as COLA, STS-B, and RTE, where accuracy gains reached 80.60%, 74.37%, and 83.80% respectively. These results demonstrate that the sample-level variational approach effectively captures task-specific features across different modalities, making it a flexible and powerful method for model merging.

**Table 1:** Multi-task performance when merging ViT-B/32 models on eight visual tasks.

| Methods | Cars | DTD | EuroSAT | GTSRB | MNIST | RESISC45 | SUN397 | SVHN | Avg Acc |
|---|---|---|---|---|---|---|---|---|---|
| Pre-trained | 59.63 | 44.13 | 45.74 | 32.60 | 48.26 | 60.27 | 63.53 | 31.63 | 48.22 |
| Task Arithmetic [17] | 60.23 | 55.18 | 80.26 | 66.21 | 95.94 | 69.17 | 64.09 | 76.27 | 70.92 |
| Ties-Merging [47] | 65.17 | 59.43 | 78.03 | 67.55 | 97.49 | 73.29 | 76.98 | 84.70 | 75.33 |
| AdaMerging [49] | 69.90 | 50.10 | 83.40 | 82.40 | 95.70 | 73.10 | 69.80 | 87.30 | 76.46 |
| DARE [50] | 61.89 | 55.19 | 79.89 | 66.03 | 95.87 | 68.94 | 63.69 | 75.98 | 70.94 |
| aTLAS [51] | 71.59 | 78.93 | 93.05 | 88.33 | 96.87 | 87.39 | 67.94 | 85.47 | 83.70 |
| WUDI-Merging [4] | 70.88 | 71.53 | 95.74 | **93.12** | 98.01 | 86.21 | **72.35** | **92.01** | 84.98 |
| **VTVC (Ours)** | **76.02** | **86.88** | **95.84** | 90.57 | **98.24** | **91.35** | 71.19 | 89.53 | **87.45** |

**Table 2:** Multi-task performance when merging RoBERTa-base model on NLP tasks.

| Methods | COLA | STS-2 | MRPC | STS-B | QQP | QNLI | MNLI | RTE | Avg Acc |
|---|---|---|---|---|---|---|---|---|---|
| Pre-trained | 0.00 | 53.76 | 85.01 | 4.01 | 37.48 | 53.05 | 37.09 | 71.19 | 42.70 |
| Weight Averaging | 0.00 | 59.21 | 85.79 | 46.99 | 45.37 | 63.94 | 48.00 | 71.19 | 52.56 |
| Task Arithmetic [17] | 8.35 | 88.26 | **89.57** | 32.84 | **82.03** | 85.40 | 75.54 | 80.43 | 67.80 |
| Ties-Merging [47] | 31.76 | 88.86 | 86.18 | 10.94 | 61.05 | **85.94** | **83.01** | 69.56 | 64.66 |
| DARE [50] | 0.00 | 88.14 | 86.61 | 30.19 | 84.33 | 79.09 | 63.95 | 77.16 | 63.68 |
| **VTVC (Ours)** | **80.60** | **90.60** | 88.48 | **74.37** | 80.40 | 83.20 | 74.00 | **83.80** | **81.93** |

**Performance on Task Negation.** Task negation aims to remove specific knowledge from a merged model while preserving other capabilities. The goal is to reduce performance on the target task while maintaining accuracy on control tasks. As shown in Table 3, VTVC achieves the best task negation performance with the lowest target accuracy and competitive control accuracy. This demonstrates that VTVC can selectively remove task-specific knowledge without significantly affecting other tasks. More detailed results are provided in Appendix C.1.

**Table 3:** Performance of task negation averaged across eight datasets.

| Methods | Target($\downarrow$) | Control($\uparrow$) |
|---|---|---|
| Task Arithmetic | 22.62 | 60.73 |
| Ties-Merging | 20.93 | **61.52** |
| TaLoS | 17.28 | 61.33 |
| aTLAS | 19.42 | 61.31 |
| **VTVC (Ours)** | **16.75** | 60.84 |

## 5.3 Ablation Studies and Analysis

In this section, we analyze the contribution of each component through ablation experiments. All ablation experiments are conducted under a unified setting. To avoid the impact of data augmentation strategies and improve computational efficiency, we use pre-computed feature representations with a static data augmentation strategy.

**Benefit of sample-specific task vectors.** Sample-specific task vector composition consistently outperforms task-level baselines, as shown in Table 4. The improvements are especially clear on Cars, SVHN, and GTSRB datasets. These datasets show greater diversity within the sample and differences between domains. SVHN contains data features that differ significantly from pre-training source data. GTSRB includes various types of traffic signs. Cars features different automobile models and viewing angles. These characteristics make unified task-level vectors insufficient for capturing sample-specific differences. Our approach calculates combination coefficients for each input sample, enabling more accurate and adaptive knowledge integration. As a result, improvements are greatest on datasets with high variability and notable domain gaps. This confirms the effectiveness of our sample-specific task vectors when handling complex data scenarios. All results are based on features encoded by the ViT-B/32 model. More detailed results are provided in Appendix C.2.

**Benefit of variational composition of task vectors.** Table 4 shows that the variational inference framework enhances the overall performance of the model. This is particularly evident in datasets with stronger differences between domains, such as EuroSAT, MNIST, and SVHN. This improvement is attributed to explicit uncertainty modeling of task vector coefficients. This provides better robustness when processing cross-domain data variations. The KL divergence regularization effectively reduces overfitting risk, improving model generalization. Deterministic methods perform slightly better on DTD, likely due to the regularity of its texture features. In this case, deterministic inference

**Table 4: Performance comparison of task-level versus sample-specific methods with deterministic and probabilistic task vectors across eight datasets.** Results demonstrate that sample-specific methods consistently outperform task-level methods across all datasets, with the variational inference framework further enhancing model performance.

| Method | Cars | DTD | EuroSAT | GTSRB | MNIST | RESISC45 | SUN397 | SVHN | Average |
|---|---|---|---|---|---|---|---|---|---|
| Pre-trained | 58.56% | 42.66% | 40.74% | 26.45% | 25.23% | 54.60% | 46.46% | 9.25% | 37.99% |
| *Deterministic-based* | | | | | | | | | |
| Task-level [51] | 62.99% | 73.29% | 92.67% | 81.11% | 97.13% | 88.41% | 67.16% | 58.01% | 77.60% |
| **Sample-specific** | 74.75% | **85.97%** | 96.07% | 85.90% | 98.07% | 90.81% | 68.01% | 67.17% | 83.34% |
| *Probabilistic-based* | | | | | | | | | |
| Task-level | 73.06% | 80.64% | 96.19% | 85.66% | 98.65% | 90.00% | 67.95% | 64.23% | 82.05% |
| **Sample-specific** | **75.33%** | 85.71% | **96.33%** | **86.76%** | **98.72%** | **90.98%** | **72.97%** | **67.78%** | **84.32%** |

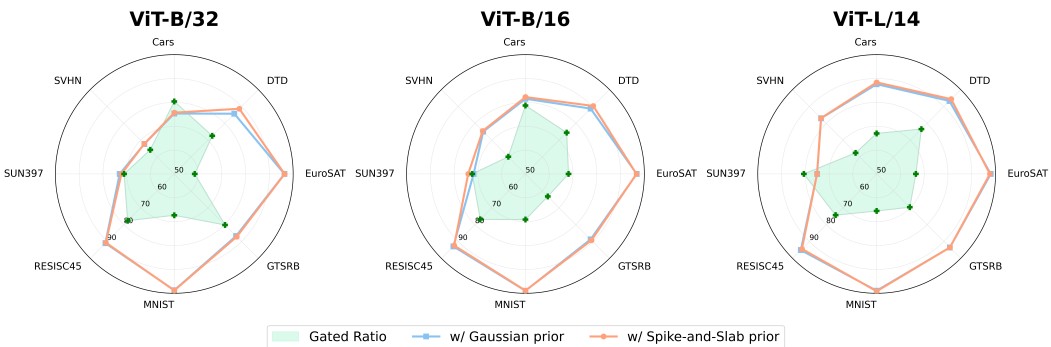

**Figure 4: Comparison of Accuracy and Gated Ratio under Gaussian and Spike-and-Slab Priors.** We visualize model accuracy and gated ratios for three ViT architectures across eight image classification datasets. The Spike-and-Slab prior consistently delivers higher accuracy while activating fewer task coefficients than the Gaussian prior, demonstrating more efficient and selective task vector composition.

adequately captures the necessary information. The uncertainty of the additional parameter from variational inference may introduce minor interference. The results validate that our variational inference framework is particularly effective for heterogeneous and cross-domain scenarios. More detailed results are provided in Appendix C.3.

**Normal Gaussian vs. Spike-Slab priors.** Figure 4 compares the performance of Gaussian and Spike-and-Slab priors across different ViT models. Spike-and-Slab priors achieve substantial performance gains on most datasets. These improvements are most evident in the datasets DTD, MNIST, and SVHN, which feature complex patterns or significant cross-domain differences. Spike-and-Slab priors excel at capturing sparse structures in combination with task vectors. They guide the model to select only the most relevant dimensions for each task. The gated ratio clearly demonstrates this effect. The increased sparsity reduces interference from redundant information and improves both efficiency and performance. It also reduces computational costs during inference. On EuroSAT and RESISC45, Gaussian priors perform slightly better, possibly because these tasks benefit from dense, high-dimensional feature representations. Their tasks likely require comprehensive feature combinations across dimensions. Spike-and-Slab priors perform exceptionally well on datasets with complex and heterogeneous characteristics. The sparse approach improves cross-domain performance and makes the composition of task vectors more efficient. More detailed results are provided in Appendix C.3.

**Effect of controllable posterior with gated sampling.** Table 5 evaluates the effectiveness of the gated sampling mechanism. Introducing the gated sampling leads to a clear overall performance improvement over the baseline Spike-Slab prior. This improvement is particularly significant on datasets such as DTD, EuroSAT, and RESISC45. The gated sampling effectively identifies and preserves critical task knowledge while suppressing noise and unreliable components by evaluating the reliability of composition coefficients. The controllable posterior with gated sampling yields a more stable and interpretable component selection compared to inference with random sampling. Analysis in different data sets reveals that gated sampling performs more prominently on datasets with complex and variable feature spaces, further confirming its superiority in handling heterogeneous data. These experimental results validate the effectiveness of our theoretical framework, demonstrating the important role of uncertainty-guided adaptive component selection in enhancing model generalization

**Table 5: Analysis of the adaptive gated sampling.** The gated sampling significantly improves model generalization capabilities across all datasets by selectively filtering reliable composition coefficients.

| Method | Cars | DTD | EuroSAT | GTSRB | MNIST | RESISC45 | SUN397 | SVHN | Average |
|---|---|---|---|---|---|---|---|---|---|
| w/o Gated Sampling | 75.69% | 88.66% | 96.41% | **87.17%** | 98.81% | 90.71% | 72.25% | 67.85% | 84.69% |
| **w/ Gated Sampling** | **77.78%** | **93.31%** | **97.04%** | 87.16% | **98.91%** | **92.08%** | **73.47%** | **68.57%** | **86.04%** |

capability and robustness. All results are based on features encoded by the ViT-B/32 model. More detailed results are provided in Appendix C.3. To further verify the effectiveness and transferability of the controllable posterior, we apply the gating module as a filter on the aTLAS method. The evaluation shows performance improvements across all datasets and three ViT architectures. Results are provided in Appendix C.4.

**Relationship Between Model Scale and Gated Ratio.** Figure 5 illustrates the impact of gated sampling on average accuracy and activation ratios across three Vision Transformer (ViT) models of increasing scale: ViT-B/32, ViT-B/16, and ViT-L/14. The results clearly demonstrate significant

accuracy improvements when incorporating gated sampling in all models. The improvements are most pronounced in larger models (ViT-L/14), highlighting the scalability advantage of our method. Notably, as the model scale increases, the activation coefficient ratio with gated posterior rises significantly, while models without gated sampling maintain relatively stable or slightly decreasing activation ratios. This pattern suggests conventional variational methods struggle to accommodate the knowledge integration demands of increasingly complex models. In contrast, our gated sampling dynamically adjusts knowledge integration strategies based on the model scale. It preserves more valuable task vectors in larger models, enabling more efficient and precise model editing. This phenomenon is consistent with the increased representational capacity and flexibility of larger models, which benefit more from adaptive gating strategies.

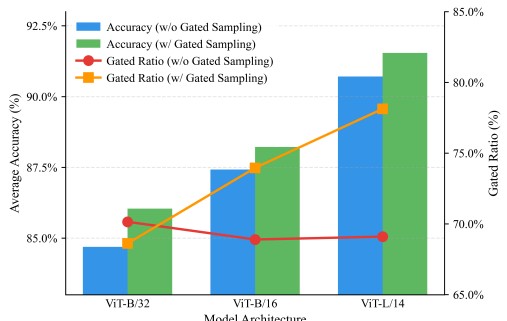

**Figure 5: Comparison of Accuracy and Gated Ratio Across Model Architectures.** The gated sampling consistently improves prediction accuracy across all models. As model complexity increases, our framework retains more coefficients, showing the effectiveness of gated sampling in handling more complex tasks.

## 6 Conclusion

In this paper, we proposed a new framework, Variational Task Vector Composition (VTVC), which formulates the composition process as a variational inference problem. Our Bayesian approach enables precise, sample-specific modeling that effectively captures domain-specific knowledge for each input. By introducing a Spike-and-Slab prior, our method automatically selects important components in task vectors while removing redundancy. This concept inspired our gated sampling mechanism, which further enhances reliability by deterministically and efficiently selecting highly informative task vectors based on uncertainty. Extensive experiments on both vision and language benchmarks demonstrate that VTVC significantly improves performance on various operations, including task addition and task negation. Overall, our framework offers a new, probabilistic perspective on task arithmetic and broadens the application potential of task vectors for adaptive model editing.

**Limitations and Future Work.** A key direction for future research is improving the offline training efficiency and scalability of our framework. While VTVC enables fast online inference, it relies on a computationally intensive stage to train the inference network, especially as the number of tasks and model scale increase.

**Acknowledgements.** This work was partially supported by the National Natural Science Foundation of China under Grant No. 62176068.

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

# A Mathematical Derivation of Variational Composition of Task Vectors

## A.1 Derivation of the ELBO Objective Function

In the variational composition of task vectors, our main objective is to infer the posterior distribution $p(\mathbf{z}|\mathbf{x}, \mathbf{y})$, where $\mathbf{z}$ denotes the composition coefficients of task vectors, $\mathbf{x}$ denotes the input data, and $\mathbf{y}$ represents the labels. According to Bayes' theorem, this posterior is given by:

$$p(\mathbf{z}|\mathbf{x}, \mathbf{y}) = \frac{p(\mathbf{y}|\mathbf{x}, \mathbf{z})p(\mathbf{z})}{p(\mathbf{y}|\mathbf{x})} \tag{15}$$

where the marginal likelihood $p(\mathbf{y}|\mathbf{x})$ is defined as:

$$p(\mathbf{y}|\mathbf{x}) = \int p(\mathbf{y}|\mathbf{x}, \mathbf{z})p(\mathbf{z})d\mathbf{z} \tag{16}$$

Since directly computing the above integral is typically intractable, we introduce a variational approximate posterior distribution $q(\mathbf{z}|\mathbf{x})$ to approximate the true posterior. We derive a lower bound on the log marginal likelihood:

$$\log p(\mathbf{y}|\mathbf{x}) = \log \int p(\mathbf{y}|\mathbf{x}, \mathbf{z})p(\mathbf{z})d\mathbf{z} \tag{17}$$

Introducing the variational distribution $q(\mathbf{z}|\mathbf{x})$:

$$\begin{aligned}
\log p(\mathbf{y}|\mathbf{x}) &= \log \int p(\mathbf{y}|\mathbf{x}, \mathbf{z})p(\mathbf{z})d\mathbf{z} \\
&= \log \int \frac{p(\mathbf{y}|\mathbf{x}, \mathbf{z})p(\mathbf{z})}{q(\mathbf{z}|\mathbf{x})}q(\mathbf{z}|\mathbf{x})d\mathbf{z}
\end{aligned} \tag{18}$$

Since the logarithm is a concave function, by Jensen's inequality:

$$\begin{aligned}
\log p(\mathbf{y}|\mathbf{x}) &= \log \mathbb{E}_{q(\mathbf{z}|\mathbf{x})}\left[\frac{p(\mathbf{y}|\mathbf{x}, \mathbf{z})p(\mathbf{z})}{q(\mathbf{z}|\mathbf{x})}\right] \\
&\geq \mathbb{E}_{q(\mathbf{z}|\mathbf{x})}\left[\log \frac{p(\mathbf{y}|\mathbf{x}, \mathbf{z})p(\mathbf{z})}{q(\mathbf{z}|\mathbf{x})}\right] \\
&= \mathbb{E}_{q(\mathbf{z}|\mathbf{x})}[\log p(\mathbf{y}|\mathbf{x}, \mathbf{z})] + \mathbb{E}_{q(\mathbf{z}|\mathbf{x})}\left[\log \frac{p(\mathbf{z})}{q(\mathbf{z}|\mathbf{x})}\right] \\
&= \mathbb{E}_{q(\mathbf{z}|\mathbf{x})}[\log p(\mathbf{y}|\mathbf{x}, \mathbf{z})] - D_{\mathbf{KL}}(q(\mathbf{z}|\mathbf{x})||p(\mathbf{z}))
\end{aligned} \tag{19}$$

This derives Equation 4 from the paper:

$$\log p(\mathbf{y}|\mathbf{x}) \geq \mathbb{E}_{q(\mathbf{z}|\mathbf{x})}[\log p(\mathbf{y}|\mathbf{x}, \mathbf{z})] - D_{\mathbf{KL}}(q(\mathbf{z}|\mathbf{x})||p(\mathbf{z})) \tag{20}$$

This inequality's right-hand side is known as the Evidence Lower BOund (ELBO). It comprises two critical components. The first term $\mathbb{E}_{q(\mathbf{z}|\mathbf{x})}[\log p(\mathbf{y}|\mathbf{x}, \mathbf{z})]$ represents the expected logarithmic likelihood, which guides the model toward accurate predictions. The second term $D_{\mathbf{KL}}(q(\mathbf{z}|\mathbf{x})||p(\mathbf{z}))$ is the KL divergence term, regularizing the posterior toward the prior distribution, thus preventing overfitting.

Maximizing ELBO is equivalent to minimizing the KL divergence between the variational posterior and the true posterior. Through optimization of the ELBO, we can learn optimal composition coefficients for task vectors while maintaining robust generalization capabilities.

## A.2 Derivation of ELBO with Spike-and-Slab Prior

In the Spike-and-Slab prior based variational framework, we introduce binary indicator variables $\omega$ to determine which composition coefficients are retained and which are set to zero. We define a joint variational posterior distribution $q_\Phi(\omega, \mathbf{z}|\mathbf{x})$, where $\Phi$ represents neural network parameters. The ELBO can be expressed as:

$$\mathcal{L}_{\mathbf{ELBO}} = \mathbb{E}_{q_\Phi(\omega, \mathbf{z}|\mathbf{x})}[\log p(\mathbf{y}|\mathbf{x}, \omega, \mathbf{z})] - D_{\mathbf{KL}}(q_\Phi(\omega, \mathbf{z}|\mathbf{x})||p(\omega, \mathbf{z})) \tag{21}$$

Using conditional probability decomposition, we can factorize the joint variational posterior as:
$$q_\Phi(\omega, \mathbf{z}|\mathbf{x}) = q_\Phi(\mathbf{z}|\mathbf{x})q_\Phi(\omega|\mathbf{z}, \mathbf{x}) \tag{22}$$
Similarly, the prior distribution can be decomposed as:
$$p(\omega, \mathbf{z}) = p(\mathbf{z})p(\omega|\mathbf{z}) \tag{23}$$
Based on the above factorization, the KL divergence term can be further expanded:

$$
\begin{aligned}
D_{\mathbf{KL}}(q_\Phi(\omega, \mathbf{z}|\mathbf{x})||p(\omega, \mathbf{z})) &= D_{\mathbf{KL}}(q_\Phi(\mathbf{z}|\mathbf{x})q_\Phi(\omega|\mathbf{z}, \mathbf{x})||p(\mathbf{z})p(\omega|\mathbf{z})) \\
&= \int q_\Phi(\mathbf{z}|\mathbf{x}) \log \frac{q_\Phi(\mathbf{z}|\mathbf{x})}{p(\mathbf{z})} d\mathbf{z} \\
&\quad + \int q_\Phi(\mathbf{z}|\mathbf{x}) \left[ \int q_\Phi(\omega|\mathbf{z}, \mathbf{x}) \log \frac{q_\Phi(\omega|\mathbf{z}, \mathbf{x})}{p(\omega|\mathbf{z})} d\omega \right] d\mathbf{z} \\
&= D_{\mathbf{KL}}(q_\Phi(\mathbf{z}|\mathbf{x})||p(\mathbf{z})) + \mathbb{E}_{q_\Phi(\mathbf{z}|\mathbf{x})}[D_{\mathbf{KL}}(q_\Phi(\omega|\mathbf{z}, \mathbf{x})||p(\omega|\mathbf{z}))]
\end{aligned} \tag{24}
$$

Therefore, the ELBO can be rewritten as:
$$\mathcal{L}_{\mathbf{ELBO}} = \mathbb{E}_{q_\Phi(\omega,\mathbf{z}|\mathbf{x})}[\log p(\mathbf{y}|\mathbf{x}, \omega, \mathbf{z})] - D_{\mathbf{KL}}(q_\Phi(\mathbf{z}|\mathbf{x})||p(\mathbf{z})) - \mathbb{E}_{q_\Phi(\mathbf{z}|\mathbf{x})}[D_{\mathbf{KL}}(q_\Phi(\omega|\mathbf{z}, \mathbf{x})||p(\omega|\mathbf{z}))] \tag{25}$$

In our model, we formulate a comprehensive probabilistic framework by defining the following distribution structure: The binary variables in the variational posterior follow $q_\Phi(\omega_i^j|\mathbf{x}) = \text{Bernoulli}(\gamma_i^j(\mathbf{x}))$, where $\gamma_i^j(\mathbf{x})$ represents the inclusion probability for each coefficient, while their prior counterparts are distributed according to $p(\omega_i^j) = \text{Bernoulli}(\pi)$, with $\pi$ serving as the global sparsity parameter. For the coefficient values themselves, when $\omega_i^j = 1$, the posterior distribution is characterized by $q_\Phi(\mathbf{z}_i^j|\omega_i^j = 1, \mathbf{x}) = \mathcal{N}(\mathbf{z}_i^j; \mu_{ij}(\mathbf{x}), \sigma_{ij}^2(\mathbf{x}))$, and correspondingly, the prior follows $p(\mathbf{z}_i^j|\omega_i^j = 1) = \mathcal{N}(\mathbf{z}_i^j; 0, \sigma^2)$, establishing a complete probabilistic specification of our variable selection mechanism.

For the binary indicator variables $\omega$, the KL divergence is calculated as:

$$
\begin{aligned}
D_{\mathbf{KL}}(q_\Phi(\omega|\mathbf{x})||p(\omega)) &= \sum_{ij} D_{\mathbf{KL}}(\text{Bernoulli}(\gamma_i^j(\mathbf{x}))||\text{Bernoulli}(\pi)) \\
&= \sum_{ij} \gamma_i^j(\mathbf{x}) \log \frac{\gamma_i^j(\mathbf{x})}{\pi} + (1 - \gamma_i^j(\mathbf{x})) \log \frac{1 - \gamma_i^j(\mathbf{x})}{1 - \pi}
\end{aligned} \tag{26}
$$

When $\omega_i^j = 1$, the KL divergence for coefficient $z_i^j$ is:

$$
\begin{aligned}
D_{\mathbf{KL}}(q_\Phi(\mathbf{z}_i^j|\omega_i^j = 1, \mathbf{x})||p(\mathbf{z}_i^j|\omega_i^j = 1)) &= D_{\mathbf{KL}}(\mathcal{N}(\mathbf{z}_i^j; \mu_{ij}(\mathbf{x}), \sigma_{ij}^2(\mathbf{x}))||\mathcal{N}(\mathbf{z}_i^j; 0, \sigma^2)) \\
&= \frac{1}{2} \left[ \log \frac{\sigma^2}{\sigma_{ij}^2(\mathbf{x})} + \frac{\sigma_{ij}^2(\mathbf{x}) + \mu_{ij}^2(\mathbf{x})}{\sigma^2} - 1 \right]
\end{aligned} \tag{27}
$$

Combining the above KL divergence calculations, and considering that when $\omega_i^j = 0$, the coefficient is forced to zero (thus contributing no additional KL divergence), the complete form of the ELBO is:

$$
\begin{aligned}
\mathcal{L}_{\mathbf{ELBO}} = {}& \mathbb{E}_{q_\Phi(\omega,\mathbf{z}|\mathbf{x})}[\log p(\mathbf{y}|\mathbf{x}, \omega, \mathbf{z})] \\
& - \left[ \sum_{ij} \gamma_i^j(\mathbf{x}) \log \frac{\gamma_i^j(\mathbf{x})}{\pi} + \sum_{ij}(1 - \gamma_i^j(\mathbf{x})) \log \frac{1 - \gamma_i^j(\mathbf{x})}{1 - \pi} \right] \\
& - \sum_{ij} \gamma_i^j(\mathbf{x}) \cdot \frac{1}{2} \left[ \log \frac{\sigma^2}{\sigma_{ij}^2(\mathbf{x})} + \frac{\sigma_{ij}^2(\mathbf{x}) + \mu_{ij}^2(\mathbf{x})}{\sigma^2} - 1 \right]
\end{aligned} \tag{28}
$$

This represents the complete expansion of Equation 6. The first term is the expected log-likelihood, representing the model's predictive accuracy; the second term is the KL divergence of binary indicator variables, controlling the sparsity pattern; and the third term is the conditional KL divergence of non-zero coefficients, weighted by their inclusion probabilities, ensuring a reasonable distribution of weight values. By optimizing this objective function, we achieve adaptive sparse composition of task vectors while maintaining the model's generalization capability.

# B Regularization and Hyperparameter Settings

**Regularization Overview**. In our adaptive gating mechanism, regularization plays a crucial role in ensuring training stability and enhancing model generalization capability. Our proposed regularization strategy addresses several key challenges: uncertainty in gating decisions, insufficient parameter space exploration, and information integration efficiency. When coefficient values approach thresholds, small perturbations can cause unstable selection results; training processes tend to converge prematurely to suboptimal solutions; and selected components must effectively represent sample features. The total regularization objective $\mathcal{L}_{reg}(\mathbf{z}, \mathcal{U}; \Psi)$ comprises multiple carefully designed components, each aiming to jointly improve the robustness and effectiveness of the gating mechanism.

**Boundary Loss**. The boundary loss $\mathcal{L}_b$ is specifically designed to penalize coefficients near thresholds, enhancing stability by encouraging more decisive selection decisions. When coefficient values lie in ambiguous threshold regions, minor input variations can cause significant fluctuations in component selection, reducing model robustness and generalization capacity. The boundary loss is defined as:

$$\mathcal{L}_b(\mathbf{z}, \Psi) = \sum_{i=1}^{n} \sum_{j=1}^{b} \max(0, m - |\mathbf{z}_{i,j} - \Gamma(U_{i,j}; \Psi)|) \tag{29}$$

This loss function imposes a positive penalty when the absolute difference between coefficient $\mathbf{z}_{i,j}$ and threshold $\Gamma(U_{i,j}; \Psi)$ falls below boundary width $m$, otherwise zero, effectively pushing coefficients away from unstable threshold regions to form more decisive binary selection patterns. In our experiments, the default boundary width parameter $m = 0.1$ effectively balances clear decision-making with model flexibility.

**Exploration Loss**. Exploration loss $\mathcal{L}_e$ promotes threshold-related parameters ($\psi_1$ and $\psi_2$) to explore broader parameter spaces, preventing training from converging to local optima. By encouraging these parameters to deviate from their initial values, the model explores potentially superior parameter configurations, enhancing the adaptability of the gating mechanism. The exploration loss is defined as:

$$\mathcal{L}_e(\Psi) = -\log\left(|\psi_1 - \psi_1^0| + \epsilon\right) - \log\left(|\psi_2 - \psi_2^0| + \epsilon\right) \tag{30}$$

This loss function decreases as parameters deviate from their initial configurations by taking the negative logarithm of the parameter-initial value difference, thus incentivizing broader parameter space exploration. Here, $\epsilon$ represents a small constant (typically $10^{-5}$) that ensures numerical stability. The uncertainty loss $\mathcal{L}_u$ is defined as:

$$\mathcal{L}_u(\mathcal{U}) = \sum_{i=1}^{n} \sum_{j=1}^{b} \mathcal{U}_{i,j} \cdot \mathbb{I}[\mathbf{z}_{i,j} \neq 0] \tag{31}$$

This loss penalizes selected (non-zero) coefficients with high uncertainty, encouraging the model to reduce uncertainty while retaining coefficients, further enhancing gating decision reliability. The comprehensive regularization objective integrates these components:

$$\mathcal{L}_{reg}(\mathbf{z}, \mathcal{U}; \Psi) = \lambda_b \mathcal{L}_b(\mathbf{z}, \Psi) + \lambda_e \mathcal{L}_e(\Psi) + \lambda_u \mathcal{L}_u(\mathcal{U}) \tag{32}$$

**Hyperparameter Settings and Sensitivity Analysis**. In our experimental implementation, we used a set of meticulously tuned hyperparameters: boundary width $m = 0.1$, boundary loss weight $\lambda_b = 10^{-4}$, exploration loss weight $\lambda_e = 10^{-3}$, uncertainty loss weight $\lambda_u = 10^{-2}$, base threshold initial value $\psi_1^0 = 0.05$, uncertainty sensitivity initial value $\psi_2^0 = 1.0$, global regularization coefficient $\lambda = 10^{-3}$ (Equation 13), and gating temperature parameter $\rho = 20.0$ (Equation 10). Sensitivity analysis indicates that model performance is relatively robust to variations in $\lambda_b$ within the range $[10^{-5}, 10^{-3}]$; more sensitive to $\lambda_e$, where increasing this value ($> 10^{-2}$) promotes more aggressive parameter exploration but can lead to training instability, while decreasing it ($< 10^{-4}$) can cause the model to converge to local optima. Performance is optimal when $\lambda_u$ is within $[5 \times 10^{-3}, 5 \times 10^{-2}]$. The initial setting of the base threshold $\psi_1^0$ significantly impacts model performance: excessive values ($> 0.1$) lead to overly sparse representations, compromising information retention; insufficient values ($< 0.01$) retain too many redundant components, increasing overfit risk. For different feature dimensions and dataset characteristics, we recommend setting $\psi_1^0$ within the range $[0.03, 0.08]$ and automatically selecting optimal configurations through validation sets. Comprehensive experimental results demonstrate that our proposed regularization strategy and parameter configurations significantly enhance the effectiveness of the adaptive gating mechanism, enabling reliable component selection across diverse task vector spaces.

# C Supplementary Experimental Results

## C.1 Detailed results of task negation

**Table 6:** Task negation on ViT-B/32 across 8 datasets. Lower is better for Target (Tgt.) and higher is better for Control (Ctr.).

| | Cars | | DTD | | EuroSAT | | GTSRB | | MNIST | | RESISC45 | | SUN397 | | SVHN | | Average | |
|---|---|---|---|---|---|---|---|---|---|---|---|---|---|---|---|---|---|---|
| | Tgt. | Ctr. | Tgt. | Ctr. | Tgt. | Ctr. | Tgt. | Ctr. | Tgt. | Ctr. | Tgt. | Ctr. | Tgt. | Ctr. | Tgt. | Ctr. | Tgt. | Ctr. |
| Task Arithmetic [17] | 35.47 | 60.73 | 26.96 | 60.67 | 11.59 | 60.67 | 7.17 | 60.40 | 12.70 | 60.86 | 32.08 | **61.23** | 47.96 | 60.67 | 7.03 | 60.60 | 22.62 | 60.73 |
| TIES-Merging [47] | 35.49 | **61.63** | 23.28 | **61.19** | 11.37 | 61.11 | 10.39 | **62.32** | 13.61 | 62.63 | 21.76 | 60.82 | 43.71 | 60.48 | 7.82 | 61.94 | 20.93 | **61.52** |
| TaLoS [19] | 28.30 | 60.88 | **18.99** | 59.76 | 9.63 | **62.13** | 5.20 | 62.01 | 12.18 | **63.00** | **13.56** | 60.27 | **41.87** | 60.47 | 8.49 | 62.10 | 17.28 | 61.33 |
| aTLAS [51] | 28.95 | 60.52 | 25.21 | 60.48 | 10.44 | 61.62 | 5.51 | 60.67 | 10.76 | 62.90 | 20.95 | 60.72 | 46.29 | **60.82** | 7.28 | **62.72** | 19.42 | 61.31 |
| **VTVC (Ours)** | **25.48** | 60.13 | 21.19 | 59.98 | **9.26** | 61.59 | **4.18** | 60.74 | **8.57** | 62.83 | 15.95 | 60.36 | 42.66 | 60.27 | **6.74** | 60.84 | **16.75** | 60.84 |

## C.2 Detailed results of task-level vs. sample-specific and deterministic vs. probabilistic

**Table 7: Detailed accuracy comparison (%) across ViT models and datasets.** We report zero-shot, deterministic (task-level and sample-specific), and probabilistic (task-level and sample-specific) performances for each model. Highest performance in each dataset is highlighted in **bold**.

| Method | Cars | DTD | EuroSAT | GTSRB | MNIST | RESISC45 | SUN397 | SVHN | Avg. |
|---|---|---|---|---|---|---|---|---|---|
| *ViT-B/32* | | | | | | | | | |
| Pre-trained | 58.56 | 42.66 | 40.74 | 26.45 | 25.23 | 54.60 | 46.46 | 9.25 | 37.99 |
| *Deterministic-based* | | | | | | | | | |
| Task-level | 62.99 | 73.29 | 92.67 | 81.11 | 97.13 | 88.41 | 67.16 | 58.01 | 77.60 |
| Sample-specific | 74.75 | **85.97** | 96.07 | 85.90 | 98.07 | 90.81 | 68.01 | 67.17 | 83.34 |
| *Probabilistic-based* | | | | | | | | | |
| Task-level | 73.06 | 80.64 | 96.19 | 85.66 | 98.65 | 90.00 | 67.95 | 64.23 | 82.05 |
| Sample-specific | **75.33** | 85.71 | **96.33** | **86.76** | **98.72** | **90.98** | **72.97** | **67.78** | **84.32** |
| *ViT-B/16* | | | | | | | | | |
| Zero-shot | 63.45 | 43.77 | 50.63 | 35.22 | 19.29 | 55.70 | 47.13 | 15.53 | 41.34 |
| *Deterministic-based* | | | | | | | | | |
| Task-level | 71.61 | 75.92 | 93.56 | 82.15 | 97.36 | 90.37 | 69.54 | 66.67 | 80.90 |
| Sample-specific | 80.59 | **90.10** | 96.96 | 88.04 | 98.85 | 92.17 | 70.02 | 74.97 | 86.46 |
| *Probabilistic-based* | | | | | | | | | |
| Task-level | 81.32 | 82.44 | 96.19 | 88.18 | 98.93 | 91.57 | 69.95 | 70.50 | 84.87 |
| Sample-specific | **81.52** | 88.75 | **96.78** | **88.76** | **98.93** | **92.90** | 71.81 | **75.12** | **86.82** |
| *ViT-L/14* | | | | | | | | | |
| Zero-shot | 76.40 | 52.10 | 55.63 | 45.61 | 52.59 | 63.41 | 50.41 | 41.73 | 54.74 |
| *Deterministic-based* | | | | | | | | | |
| Task-level | 85.39 | 83.41 | 96.89 | 91.55 | 98.76 | 93.79 | 71.05 | 76.02 | 87.11 |
| Sample-specific | 87.29 | **96.84** | 97.91 | 93.46 | **99.09** | 94.30 | 73.90 | 82.38 | 90.65 |
| *Probabilistic-based* | | | | | | | | | |
| Task-level | 88.47 | 90.08 | 97.78 | 92.87 | 99.01 | 93.54 | 72.33 | 78.46 | 89.07 |
| Sample-specific | **89.64** | 93.28 | **97.99** | **93.55** | 98.92 | **95.11** | **75.05** | **83.03** | **90.82** |

## C.3 Detailed results of Gaussian vs. Spike-and-Slab priors

**Table 8: Performance comparison between Gaussian and Spike-and-Slab prior models (%) across ViT models and datasets.** Highest performance in each dataset is highlighted in **bold**. Gated ratio values are reported in *italics*.

| Method | Cars | DTD | EuroSAT | GTSRB | MNIST | RESISC45 | SUN397 | SVHN | Avg. |
|---|---|---|---|---|---|---|---|---|---|
| *ViT-B/32* | | | | | | | | | |
| Gaussian prior | 75.33% | 85.71% | 96.33% | 86.76% | 98.72% | **90.98%** | **72.97%** | 67.78% | 84.32% |
| Spike-and-Slab prior | **75.69%** | **88.66%** | **96.41%** | **87.17%** | **98.81%** | 90.71% | 72.25% | **67.85%** | 84.69% |
| *ViT-B/16* | | | | | | | | | |
| Gaussian prior | 81.52% | 88.75% | **96.78%** | 88.76% | 98.85% | 92.17% | 71.81% | 74.97% | 86.61% |
| Spike-and-Slab prior | **82.23%** | **90.31%** | 96.70% | **89.27%** | **98.87%** | **92.40%** | **74.10%** | **75.46%** | **87.42%** |
| *ViT-L/14* | | | | | | | | | |
| Gaussian prior | 87.29% | 93.28% | **97.67%** | 93.46% | 99.09% | 94.30% | 73.90% | 82.38% | 90.38% |
| Spike-and-Slab prior | **88.38%** | **94.35%** | 97.63% | **93.56%** | **99.14%** | **94.43%** | **75.07%** | **83.09%** | **90.71%** |

## C.4 Detailed results of gated sampling

**Table 9: Accuracy and gating ratio comparison (%) across ViT models and datasets.** Performance and average gated ratio are reported for models with and without gated sampling. The highest performance in each dataset is highlighted in **bold**. Gated ratio values are in *italics*.

| Method | Cars | DTD | EuroSAT | GTSRB | MNIST | RESISC45 | SUN397 | SVHN | Avg. |
|---|---|---|---|---|---|---|---|---|---|
| *ViT-B/32* | | | | | | | | | |
| w/o Gated Sampling | 75.69% | 88.66% | 96.41% | **87.17%** | 98.81% | 90.71% | 72.25% | 67.85% | 84.69% |
| Gated Ratio | *80.40%* | *72.60%* | *58.60%* | *80.20%* | *67.30%* | *77.70%* | *71.10%* | *64.30%* | *71.53%* |
| w/ Gated Sampling | **77.78%** | **93.31%** | **97.04%** | 87.16% | **98.91%** | **92.08%** | **73.47%** | **68.57%** | **86.04%** |
| Gated Ratio | *65.62%* | *71.88%* | *59.38%* | *71.87%* | *60.42%* | *63.54%* | *91.67%* | *64.58%* | *68.62%* |
| *ViT-B/16* | | | | | | | | | |
| w/o Gated Sampling | 82.23% | 90.31% | 96.70% | **89.27%** | 98.87% | 92.40% | **74.10%** | 75.46% | 87.42% |
| Gated Ratio | *78.70%* | *74.50%* | *68.10%* | *63.30%* | *69.10%* | *76.90%* | *72.40%* | *60.20%* | *70.40%* |
| w/ Gated Sampling | **83.61%** | **95.01%** | **97.00%** | 89.22% | **98.93%** | **93.02%** | 73.53% | **75.47%** | **88.22%** |
| Gated Ratio | *79.17%* | *73.96%* | *67.71%* | *82.23%* | *81.25%* | *77.08%* | *62.08%* | *75.21%* | *75.21%* |
| *ViT-L/14* | | | | | | | | | |
| w/o Gated Sampling | 88.38% | 94.35% | 97.63% | 93.56% | 99.14% | 94.43% | 75.07% | 83.09% | 90.71% |
| Gated Ratio | *67.00%* | *76.60%* | *66.50%* | *69.70%* | *65.50%* | *74.40%* | *62.50%* | *80.60%* | *70.35%* |
| w/ Gated Sampling | **89.45%** | **97.75%** | **97.96%** | **93.67%** | **99.15%** | **95.49%** | **75.52%** | **83.31%** | **91.54%** |
| Gated Ratio | *69.79%* | *72.92%* | *70.83%* | *85.83%* | *85.42%* | *85.42%* | *67.71%* | *78.13%* | *78.13%* |

## C.5 Applying gated sampling to aTLAS for filtering

**Table 10: Performance comparison between standard aTLAS and Gated aTLAS methods across three ViT architectures and eight datasets (%).** The gated mechanism consistently improves performance across all datasets and models. Highest performance in each dataset is highlighted in **bold**.

| Method | Cars | DTD | EuroSAT | GTSRB | MNIST | RESISC45 | SUN397 | SVHN | Avg. |
|---|---|---|---|---|---|---|---|---|---|
| *ViT-B/32* | | | | | | | | | |
| w/o Gated | 62.99% | 73.29% | 92.67% | 81.11% | 97.13% | 88.41% | 67.16% | 58.01% | 77.60% |
| w/ Gated | **71.32%** | **80.24%** | **94.96%** | **84.92%** | **97.96%** | **89.84%** | **68.33%** | **61.68%** | **81.16%** |
| *ViT-B/16* | | | | | | | | | |
| w/o Gated | 71.61% | 75.92% | 93.56% | 82.15% | 97.36% | 90.37% | 69.54% | 66.67% | 80.90% |
| w/ Gated | **80.06%** | **82.37%** | **94.96%** | **86.04%** | **98.10%** | **91.59%** | **70.29%** | **68.68%** | **84.01%** |
| *ViT-L/14* | | | | | | | | | |
| w/o Gated | 85.39% | 83.41% | 96.89% | 91.55% | 98.76% | 93.79% | 73.05% | 76.02% | 87.11% |
| w/ Gated | **86.78%** | **88.40%** | **97.33%** | **92.88%** | **98.90%** | **94.75%** | **77.19%** | **77.79%** | **88.59%** |

