# OpenReview forum: "Variational Task Vector Composition"
_NeurIPS.cc/2025/Conference — NeurIPS 2025 poster_

### Official Review · Reviewer_ZAnD · 2025-06-25

**Clarity:** 3
**Significance:** 2
**Originality:** 2
**Rating:** 4
**Confidence:** 2

**Summary:**

This paper proposes a novel framework for variational composition of learned task vectors, aiming to improve both performance and efficiency by reducing redundancy and introducing uncertainty-aware combination coefficients.

**Questions:**

1. How should we implement gated sampling for task negation?

2. If we have irrelevant task vectors, how does a spike-and-slab prior help reduce such redundancy?

**Ethical Concerns:**

["NO or VERY MINOR ethics concerns only"]

**Final Justification:**

The author has addressed my concern regarding the motivation for gated sampling.  I will keep my current rating (4 already as positive) and do not have any further questions.

**Limitations:**

Yes

**Quality:**

3

**Strengths And Weaknesses:**

Strengths

1. The structure and main contributions of this paper are clearly described. The authors provide sufficient background and prior knowledge, which helps the audience better understand the context and motivation behind this work.

2. The proposed methodology is conceptually sound and is empirically validated through experiments on real-world datasets.

Weaknesses

1. The contribution of the gated sampling mechanism appears limited, especially if similar techniques have already been introduced in other frameworks. The paper does not clearly explain any unique challenges or novel adaptations specific to applying this approach in the task vector setting.

2. There is a lack of experimental evaluation on the task unlearning scenario.

---

> ### Author Rebuttal · Authors · 2025-07-31
>
> We appreciate the reviewer’s feedback and for highlighting the clear structure and robust empirical validation of our work. Below, we address the questions and concerns.
>
> **W1: The gated sampling mechanism seems limited in contribution. Similar ideas may exist elsewhere, and the paper doesn’t explain what’s uniquely challenging or new about using it in task vector composition.**
>
> We agree that gating mechanisms have been explored in other works. However, applying such mechanisms to **task vector composition** introduces several **unique challenges** that our method addresses:
> * First, **task vectors operate in a high-dimensional parameter space**, where conventional Monte Carlo sampling leads to high variance and instability[1]. Our gated sampling mechanism avoids this by introducing a **deterministic, continuous gating function**, which allows **gradient-based optimization** while maintaining sparsity and stability.
> * Second, unlike existing gating approaches, our mechanism jointly considers **token-level uncertainty and coefficient magnitude**—a design tailored to the **uncertainty-aware and sparse nature of task composition**.
> * Finally, we show that this design improves **both efficiency and generalization**, especially when combined with Spike-and-Slab priors (see Figure 4 and Appendix C.3), which to our knowledge has **not been explored in prior task arithmetic literature**.
> We will clarify these contributions and motivations more explicitly in Section 4.3.
>
>
> **W2/Q1: Task negation**
>
> We agree that evaluating task negation is important for demonstrating the scalability and flexibility of our framework. To address this, we have added new experiments on task unlearning.
> Specifically, we introduce an **asymmetric gated sampling mechanism** that sets a lower threshold for the coefficients of target (to-be-forgotten) tasks, while preserving important components of control tasks. This allows for effective suppression of target task influence without compromising control task performance. The results on the ViT-B/32 backbone are summarized below:
>
> | Methods           | Target($\downarrow$)| Control($\uparrow$)|
> |:-|:-:|:-:|
> | Task Arithmetic[2]   | 22.62%    | 60.73%     |
> | TIES-Merging[3]      | 20.93%    | **61.52%**     |
> | TaLoS[4]             | 17.28%    | 61.33%     |
> | aTLAS[5]             | 19.42%    | 61.31%     |
> | VTVC **(Ours)**       | **16.75%**    | 60.84%     |
>
> These results show that our method achieves stronger forgetting on the target task while maintaining comparable performance on the control tasks. This demonstrates the method’s applicability to both **task addition** and **task negation** scenarios.
>
> **Q2: If we have irrelevant task vectors, how does a spike-and-slab prior help reduce such redundancy?**
>
> When task vectors contain irrelevant vectors or blocks for the current task, the **spike-and-slab prior** reduces redundancy by **pushing unimportant coefficients to exactly 0** through probabilistic modeling. This achieves structured sparsity, reduces redundancy, avoids negative transfer, and lowers computational and storage costs.
> + The spike-and-slab prior models each combination coefficient as a mixture of **point mass and Gaussian**: $p(\mathbf{z})=(1-\pi)\delta_0(\mathbf{z})+\pi\mathcal{N}(0,\sigma^2)$,
> here $\delta_0$ is the Dirac distribution at $0$ (the "spike"), and $\mathcal{N}$ is the "slab". This means the prior **prefers most coefficients to be $0$**, with only a few truly useful dimensions falling in the continuous Gaussian part. As a result, dimensions irrelevant to the current sample/task are naturally pushed to $0$ during learning, effectively "turning off" redundant task vectors or their sub-blocks.
> + In variational inference, each coefficient $\mathbf{z}_i^j$ has a binary indicator variable $\omega_i^j$: $\omega=1$ means keeping that dimension, $\omega=0$ means setting it to zero. The model uses **factorized posteriors** and jointly infers $\omega$ and $\mathbf{z}$. If a dimension contributes little to the likelihood, the optimal posterior pushes mass toward $\omega=0$ (corresponding to the prior's spike), achieving exact $0$ sparsity. The KL term in the ELBO further penalizes unnecessary non-zeros, compressing redundant dimensions.
> We hope this clarification helps address your concerns about the spike-and-slab prior.
>
> **References**:
>
> [1] Reichelt et al. Rethinking variational inference for probabilistic programs with stochastic support. NeurIPS’22.
>
> [2] Ilharco et al. Editing models with task arithmetic. ICLR’23.
>
> [3] Yadav et al. TIES-Merging: Resolving interference when merging models. NeurIPS’23.
>
> [4] Iurada et al. Efficient Model Editing with Task-Localized Sparse Fine-tuning. ICLR’25.
>
> [5] Zhang et al. Knowledge Composition using Task Vectors with Learned Anisotropic Scaling. NeurIPS’24.
>
> **Thank you again for helping us improve the paper and hope our response can resolve your concerns!**

---

> ### Comment · Reviewer_ZAnD · 2025-08-05
>
> Thank you for the rebuttal and clarification. I will keep my current rating (4 as positive) and do not have any further questions.

---

> > ### Author Response · Authors · 2025-08-07
> >
> > We sincerely thank you for your time and your positive rating. We appreciate your feedback and look forward to presenting the improved version.

---

### Official Review · Reviewer_Wknn · 2025-06-27

**Clarity:** 3
**Significance:** 3
**Originality:** 3
**Rating:** 5
**Confidence:** 2

**Summary:**

The paper proposes a method for the integration of task-specific knowledge to be generalized across tasks. The authors formulate a probabilistic model for the task vector composition task and propose a spike-and-slab variational inference framework for efficient training and sample-specific inference. The authors also propose a differentiable and deterministic gated sampling algorithm for training their model. The experiments show that their method improves the classification accuracy across datasets. The authors also empirically discussed the benefits of the gated sampling approach.

**Questions:**

What is the pretrained model in Fig 2a?
Why does MNIST show a very distinct representation but gain a lot from the knowledge integration in Table 1?
In the experiments, I understand that the image is the input and the class is the output. Do you finetune the pretrained weights for all the tasks together with the ELBO loss? If so, then coming back to my earlier point, please provide how many trainable parameters you are adding.
At the beginning of the experiments, you mentioned, “We select these datasets to evaluate the effectiveness and scalability of our approach on diverse visual and task scenarios.” I think it is incorrect to say that the approach was evaluated on diverse tasks, since only image classification was chosen.
Please clarify the difference between a deterministic and stochastic approach, and Task-level vs sample-specific?
Are the w/o Gated results in Table 2 from the same model and inference approach as the Table 1 final row? If so, then why are the numbers different?

**Ethical Concerns:**

["NO or VERY MINOR ethics concerns only"]

**Final Justification:**

I am increasing the point as the authors have clarified my doubts about the additional parameter overhead of their approach and also provided additional evidence in support of their method in NLP tasks. However, my other doubt about the incremental novelty of the author's approach vs aTLAS remains.

**Limitations:**

Yes

**Quality:**

3

**Strengths And Weaknesses:**

Strengths: The paper is clearly written and presents a good motivation through Fig 2 that there are overlaps in task-specific knowledges and a lot of redundant features. The proposed method is intuitive and easy to follow. The experiments show improved overall results for image classification across different datasets over the competitive approach. They further show accuracy improvement due to a sample-specific gated sampling approach, which was the main methodological novelty.
Weakness:
I don’t have much experience in the field of knowledge transfer through the task vector arithmetic. I enjoyed reading the paper and the main competitive approach aTLAS. I think the main limitation of this work is the limited novelty of the method over aTLAS. The method is a sparse summation of the task vectors instead of a non-sparse summation in aTLAS. Methodologically, I am not sure how much this change is novel in the ML literature.
The notations are confusing. Is z the activations from a neural network? If so, then how can it be added to the parameters in \Theta_0 in eq 12?
I liked the deterministic gated sampling algorithm, but it also brings additional parameters to be trained. How is this then beneficial to use in knowledge transfer, where the objective is to fine-une pretrained weights. If one has additional compute budget, then wouldn’t it be easier and accurate to train separate models (classifiers in this case) on separate data?

---

> ### Author Rebuttal · Authors · 2025-07-31
>
> We appreciate your constructive review and for recognizing the clarity of presentation and empirical improvements, we are glad you enjoyed reading the paper. We now offer clarifications and address the concerns raised.
>
> **W1: The main difference from aTLAS seems to be introducing sparsity into the summation. Methodologically, it’s unclear how novel this change is in the ML literature.**
>
> While our method indeed builds on aTLAS in spirit, it introduces several **non-trivial methodological departures** tailored for **sample-specific and uncertainty-aware task composition**:
>  * We go beyond fixed task-level scaling and instead propose a **variational Bayesian formulation** that **learns input-dependent coefficient distributions** rather than fixed values.
>  * Our use of **Spike-and-Slab priors** enables **structured sparsity**, which not only improves interpretability but also reduces overfitting and improves generalization on heterogeneous tasks.
> * We further design a **deterministic gating mechanism** to replace stochastic sampling, which addresses variance and scalability issues commonly encountered in variational inference over high-dimensional parameter spaces.
> * Empirically, this framework consistently outperforms aTLAS while using fewer parameters (see Fig. 4 and App. C.3), indicating that the proposed changes bring measurable benefits beyond architectural variation.
>  To our knowledge, these components—**Bayesian sample-level task vector composition with sparse gating**— have not been explored in prior work, including aTLAS. We believe this constitutes a meaningful and task-relevant extension in both the **modeling and inference** aspects.
>
>
> **W2: Notations.**
>
> Thank you for pointing out the confusion in our notation. We used $z$ in line 104 to represent the latent representation of input image $x$ from the neural network, and $\mathbf{z}$ in lines 155-156 to represent the composition coefficients in our variational framework. In Eq. 12, $\mathbf{z}(\mathbf{x}_s;\Phi)$ refers to the composition coefficients for sample $\mathbf{x}_s$, which are filtered by gating $\mathcal{G}$ (Eq. 7) and then combined with the task vector matrix $\mathcal{T}$ to produce weight increments of the same shape as $\Theta_0$, resulting in the final prediction. We will clarify the symbol definitions in our final revision to ensure a clear distinction of symbol meanings.
>
>
> **W3. Comparison of training separate models on separate data.**
>
> Regarding your concerns about additional computational costs, we provide a detailed discussion:
> + **Minimal parameter overhead**: The gated sampling model has extremely small parameter requirements. The additional parameters only include $\Psi=\lbrace \psi_1, \psi_2\rbrace $ (two scalar thresholds) and a three-layer MLP with parameters $\Phi$, totaling **<1%** of the backbone model parameters. Compared to training separate complete models for each task, the overhead of our gating mechanism is negligible. Despite this minimal cost, gating significantly improves knowledge transfer performance — achieving **1.35%** improvement over spike-and-slab prior on ViT-B/32, and **3.56%** improvement when applied to the aTLAS method. Moreover, our gated sampling method achieves state-of-the-art results across three models and eight datasets, demonstrating its excellent performance.
> + **Dynamic knowledge composition**: For your suggestion of training separate models on separate data, multi-task applications often require **dynamically switching or combining knowledge across different domains**. With separate models trained on individual datasets, it is nearly impossible to enable/suppress specific domain features at runtime. Additionally, when adding N new tasks in the future, you would need N large-scale training runs and storage for N complete model weights. In contrast, task arithmetic only requires fine-tuning once per new task and saving task vectors, with minimal storage growth during composition. Our gated sampling mechanism further improves inference efficiency and storage space, providing more flexible choices for task vector composition in multi-task scenarios.
> We hope this discussion helps you better understand the advantages of gated sampling.
>
>
> **Q1: Pre-trained model in Fig 2a.**
>
>
> The pretrained model used in Figure 2a is the CLIP-ViT-B/32 image encoder.
>
>
> **Q2: Explanation of MNIST.**
>
> You made a very careful observation. We believe the possible reason for this interesting phenomenon is as follows: In the t-SNE projection, MNIST clusters appear clearly separated from other datasets mainly because of the huge domain gap between its visual distribution (grayscale, low resolution, handwritten strokes) and natural images. Due to this difference, pretrained features alone perform poorly on MNIST. However, our sample-level composition automatically selects a few task vectors related to "contours/strokes" for handwritten digits during inference while suppressing redundant channels. Therefore, MNIST accuracy in Table 1 improves dramatically from 25.23% (using only $\Theta_0$) to 98.72%, showing the most significant gain.
>
> **Q3: Do you finetune the pretrained weights for all the tasks together with the ELBO loss? If so, then coming back to my earlier point, please provide how many trainable parameters you are adding.**
>
> In our training process, the pretrained weights are not jointly fine-tuned with the ELBO loss across all tasks. Our process consists of two stages:
> 1. **Task vector creation**: We independently fine-tune each dataset to obtain corresponding task vectors, then freeze the pretrained weights $\Theta_0$ and all $\tau$.
> 2. **Composition stage**: We only train the inference network $\Phi$(3-layer MLP) and two gating thresholds $\Psi$ using ELBO loss to predict sample-level composition coefficients $\mathbf{z}$ and perform gating.
> **Trainable parameter scale**: Using ViT-B/32 as an example, the backbone has approximately 87.8M parameters. The inference network uses a three-layer fully-connected structure, plus two scalar thresholds and a global scaling coefficient $\lambda$, totaling approximately 0.75M parameters, which is ~0.85\% of the backbone. The additional FLOPs during inference are also **<5%**. Compared to training $K$ complete separate models for $K$ tasks, our storage and update overhead is significantly reduced.
>
>
> **Q4. Single task scenario.**
>
> Thank you for pointing out the imprecise wording in our paper. You are correct – we indeed only selected image classification experiments, and we agree that expanding our method to other task scenarios is important for validating its effectiveness.
> Therefore, we have added experiments on discriminative NLP tasks. The experimental results are as follows:
>
> | Methods           | COLA | STS-2 | MRPC | STS-B | QQP | QNLI | MNLI | RTE | Avg. |
> |:-|:-:|:-:|:-:|:-:|:-:|:-:|:-:|:-:|:-:|
> | Zero-shot   | 0.00  | 53.76 |85.01 |4.01 |37.48 |53.05 |37.09 |71.19 |42.70 |
> | Weight Averaging   | 0.00  | 59.21 |85.79 |46.99 |45.37 |63.94 |48.00 |71.19 |52.56 |
> | Task Arithmetic[1]  | 8.35  | 88.26 |**89.57** |32.84 |82.03|85.40 |75.54 |80.43 |67.80 |
> | TIES-Merging[2]   | 31.76  | 88.86 |86.18 |10.94 |61.05 |**85.94** |**83.01** |69.56 |64.66 |
> | DARE[3]  | 0.00  | 88.14 |86.61 |30.19 |**84.33** |79.09 |63.95 |77.16 |63.68 |
> | VTVC **(Ours)** | **80.60**  | **90.60** |88.48 |**74.37** |80.40 |83.20 |74.00 |**83.80** |**81.93** |
>
> For discriminative language tasks, we use RoBERTa[4] as the backbone and evaluate on the 8-task GLUE benchmark[5]. We will provide detailed experimental settings and more comprehensive results in the revised version.
>
>
> **Q5. Clarify the difference between a deterministic and stochastic approach, and Task-level vs sample-specific.**
>
>
> + **Deterministic vs Stochastic approaches**: Deterministic methods directly provide a single point estimate with unique composition results. Stochastic methods treat coefficients as random variables, explicitly model uncertainty, and complete composition through distributions or sampling.
> + **Task-level vs Sample-specific approaches**: Task-level methods use coefficients that only depend on the overall "task" label, shared by all samples. Sample-specific methods use coefficients that are further influenced by specific inputs, enabling fine-grained adaptation to differences within samples.
>
>
> **Q6. Are the w/o Gated results in Table 2 from the same model and inference approach as the Table 1 final row?**
>
> Table 2 compares method performance under sparse conditions. The w/o Gated results in Table 2 come from the model with spike-and-slab prior, while the final row in Table 1 comes from a sample-level probabilistic model under Gaussian sampling. The difference between these two methods is that **the former uses a spike-and-slab prior to reduce redundancy**. Detailed results can be found in Appendix C.2.
>
>
> **References**:
>
> [1] Ilharco et al. Editing models with task arithmetic. ICLR’23.
>
> [2] Yadav et al. TIES-Merging: Resolving Interference When Merging Models. NeurIPS’23.
>
> [3] Yu et al. Language Models are Super Mario: Absorbing Abilities from Homologous Models as a Free Lunch. ICML’24.
>
> [4] Liu et al. Roberta: A robustly optimized bert pretraining approach. CoRR’19.
>
> [5] Wang et al. Glue: A multi-task benchmark and analysis platform for natural language understanding. EMNLP Workshop BlackboxNLP’18.
>
> **Thank you again for helping us improve the paper and hope our response can resolve your concerns!**

---

> > ### Comment · Reviewer_Wknn · 2025-08-05
> >
> > I thank the authors for their detailed response, which has helped me clarify the additional parameter overhead question I had, among others. I also appreciate the authors for running additional experiments on the NLP task vector composition task, which indeed show good results for their approach. Given the additional evidence in favor of their method, I am increasing my score. However, my other doubt about the incremental novelty of the author's approach vs aTLAS remains, but I agree that their adaptation of spike-and-slab variational inference and the gated sampling approach is very interesting.

---

> > > ### Author Response · Authors · 2025-08-05
> > >
> > > We sincerely thank you for recognizing our work and increasing your score. We will include the experiments we conducted during the rebuttal period in the revised manuscript. We will also clarify the difference between our work and aTLAS in the related work section.

---

### Official Review · Reviewer_Xxkm · 2025-06-28

**Clarity:** 2
**Significance:** 3
**Originality:** 2
**Rating:** 4
**Confidence:** 3

**Summary:**

The paper proposes a variational approach for task vector composition. Given a pre-trained model and target tasks, the main idea consists of sampling the weights’ difference to be added to the pre-trained weights to deliver a stronger set of weights that adapt to multiple downstream tasks. As a baseline, a simple Gaussian parameterization to the posterior is introduced, and then a spike-and-slab prior to enforce sparsity in the parameters to be updated is introduced, to enable task-specific parameter selection. Finally, the paper introduces a gating approach to dealing with the high-dimensionality of the weights and the problems it induces in Monte-Carlo sampling. The experimental setup follows prior work, showing results across 8 datasets, and 3 different ViT models.

**Questions:**

My questions are all included in the Strengths and Weaknesses section.

**Ethical Concerns:**

["NO or VERY MINOR ethics concerns only"]

**Final Justification:**

The rebuttal has addressed my concerns and provided the other reviewers' comments and consideration for the paper, I am inclining towards accepting the paper. I therefore keep my original rating, hoping for the reviewers to address the gaps in the final version.

**Limitations:**

The authors mention the high GPU demand as a limitation of their approach, although I would like to see, as I expressed before, a proper breakdown on that limitation. Other than that limitations are properly addressed.

**Quality:**

2

**Strengths And Weaknesses:**

Strengths:

The paper extends the prior work of aTLAS ([42] Zhang et al “Knowledge Composition using Task Vectors with Learned Anisotropic Scaling”, NeurIPS’24) by reformulating the task arithmetic in a variational manner. Using this approach, the task vector can be obtained directly in a sample-specific manner, showing that this approach results in better performance than when using task-specific vectors.

The main idea of the paper is interesting, well-motivated and developed. Although it requires some extra effort to match the pieces, the paper follows a proper story with a clear idea, that is, to have a sampling-based approach, uncertainty-aware, to producing task-specific vectors that enable the same neural network to deliver strong results without the need of retraining in downstream tasks.


Weaknesses:

The first shortcoming of the paper is its writing, with loose narrative and confusing notation. For instance, the authors refer first to z as the latent representation of a CLIP model, to use it immediately after to refer to the task vector coefficients. The paper needs to be polished with proper notation and definitions presented forehand. For me to understand what the authors were trying to accomplished, I had to go read first the aTLAS paper. The paper should be a bit more self-contained.

While the contribution is neat, it is not innovative; applying a Bayesian approach to an existing pipeline is a recurrent idea, and considering that it in this case it comes with a marginal advantage over existing alternatives, it is hard to ponder the contribution of the paper.

I am a bit concerned in regards to the experiments. I find the experimental setup a bit ill-defined. In particular, I miss the following details:
-	The training details: while the authors comment the datasets they use, they barely refer to the “standard benchmarks”. They depart from the pre-trained ViT models and my understanding is that they try to learn the network for parameter sampling based on the corresponding datasets. What it is not clear to me is if they trained a single model using all training partitions from the corresponding datasets, or if they trained a single model for each dataset using the corresponding trainining partitions. In the paper it is suggested at some point that gathering a strong set of task vectors to form a single model can result in strong results, however it doesn’t seem to be the case in this scenario. May the authors comment on that? It would be good to see a one-leave-out set of experiments where possible, considering that the model merely consists of a CLIP vision encoder.

-	A comparison against prior work like aTLAS is missing.

-	Table 1 shows that having a sample-specific approach clearly outperforms having a task-specific approach. However, the difference between the deterministic and probabilistic approaches is quite marginal, which is concerning.


-	The same concern shows in Figure 4: the difference between using a simple Gaussian prior w.r.t. using a Spike-and-Slab prior seems marginal to me, putting under question what the contribution of the latter is in the whole framework. Is it indeed a necessary prior?

-	What is the computational complexity of having a per-sample task vector against having task-specific weights? I understand that the former approach will incur in multiple inference steps that are not necessary in the latter approach. A tradeoff or balance between gain and complexity should be also included in the experimental section.

---

> ### Author Rebuttal · Authors · 2025-07-31
>
> Thank you for your thoughtful feedback and for noting the clarity of our research motivation and approach. In what follows, we respond to the raised points with clarifications.
>
> **W1: The paper suffers from loose narrative and confusing notation, particularly the use of **z**. It also relies too much on external papers like aTLAS to be understood.**
>
> We appreciate the reviewer’s feedback and agree that clarity can be improved. In the revised version:
> * We will  **consistently reserve** $\mathbf{z}$ for task vector coefficients and avoid reusing it for CLIP embeddings (which are now denoted as $h$).
> * We will and a **notation table** and clarify key terms early in Section 3 to make the paper more self-contained.
> * A brief summary of task composition and aTLAS will be included to reduce reliance on external papers.
> These changes aim to resolve the ambiguity and improve readability.
>
> **W2: The Bayesian angle feels like a standard extension and offers only marginal improvement, making the contribution harder to assess.**
>
> We appreciate the reviewer’s positive overall evaluation. While we adopt a Bayesian perspective, our motivation is grounded in the specific **challenge of composing task vectors under uncertainty and redundancy**, which is **not addressed in prior works** like aTLAS. Unlike previous methods that rely on deterministic, task-level composition, we introduce a **structured variational framework with Spike-and-Slab priors**, enabling **sample-specific sparse inference** and **controllable coefficient selection**.
>
> This design is **particularly suited for multi-task transfer with heterogeneous tasks**, where a fixed combination often fails to generalize. Moreover, our **deterministic gating mechanism** avoids the instability of Monte Carlo sampling in high-dimensional weight space—an aspect rarely considered in prior literature. These ingredients not only bring **consistent performance improvements**, but also offer better **sparsity-efficiency trade-offs** and **interpretability**. We hope this clarifies that our contribution is more than a marginal extension.
>
> **W3a: Standard benchmarks.**
>
> We clarify the task vector acquisition process. We use CLIP-ViT-B/32 as the base model and perform lightweight fine-tuning separately on 8 **public standard datasets** (Cars, DTD, EuroSAT, GTSRB, MNIST, RESISC45, SUN397, SVHN), which are the most commonly used visual benchmarks in aTLAS and Task-Vector literature. The fine-tuning settings are consistent with aTLAS, so each dataset corresponds to one task vector.
>
>
>
> **W3b: One-leave-out set of experiments.**
>
> Regarding your question, we have conducted one-leave-out experiments on the ViT-B/32 model(Deterministic-based+task-level), with results shown in the table below:
>
> | w/o Cars | w/o DTD | w/o EuroSAT | w/o GTSRB | w/o MNIST | w/o RESISC45 | w/o SUN397 | w/o SVHN | **Avg.** |
> |:-:|:-:|:-:|:-:|:-:|:-:|:-:|:-:|:-:|
> | 73.85  | 74.36 |71.77 |73.89 |70.59 |72.47 |73.04 |72.55 |72.78 |
>
> The experimental results show that when excluding one dataset, the average accuracy of model merging on the remaining seven datasets decreased by **4.82%** compared to merging eight models. This experimentally demonstrates that expanding the task vector pool can enhance model merging performance. For theoretical supplementation and proof, please refer to Ortiz-Jimenez et al.[5] and Li et al.[6].
>
>
> **Q1: A comparison against prior work like aTLAS is missing.**
>
> The "Deterministic-based+Task-level" results in Table 1 of our paper are actually the aTLAS results. We did not explicitly label them as "aTLAS" when writing to distinguish between task-level and sample-level effects, which may have caused confusion in understanding the comparison. We will clearly mark the aTLAS name in the final version and have added comparisons with more prior work including Task Arithmetic[1], AdaMerging[2], and WUDI-Merging[3]. The results are shown in the table below:
>
> | Methods | Cars | DTD | EuroSAT | GTSRB | MNIST | RESISC45 | SUN397 | SVHN | Avg. |
> |-|:-:|:-:|:-:|:-:|:-:|:-:|:-:|:-:|:-:|
> | Zero-shot   | 59.63  | 44.13 |45.74 |32.60 |48.26 |60.27 |63.53 |31.63 |48.22 |
> | Task Arithmetic[1]   |60.23  | 55.18 |80.26 |66.21 |95.94 |69.17 |64.09 |76.27 |70.92 |
> | AdaMerging[2]  | 56.90  | 50.10 |83.40 |82.40 |95.70|73.10 |60.80 |87.30 |73.71 |
> | WUDI-Merging[3]  | 70.88  | 71.53 |95.74 |**93.12** |98.01 |86.21 |**72.35** |**92.01** |84.98 |
> | VTVC **(Ours)** | **76.02**  | **86.88** |**95.48** |90.57 |**98.24**|**91.35** |71.19 |89.53 |**87.45** |
>
>
> **Q2: The difference between the deterministic and probabilistic approaches.**
>
> Thank you for pointing out the superiority of sample-specific approaches in our method. For the probabilistic approach, we can see from Table 1 that there is a **4.45%** improvement at the task level and a **0.98%** improvement at the sample level. However, we notice that on the most challenging dataset SUN397, the ViT-B/32 model shows a **4.96%** improvement with the sample-level probabilistic method compared to the deterministic method.
>
> We believe one reason for the smaller average improvement at the sample level is that the sample-specific method already fully utilizes each sample's features through learning a lightweight network. However, the probabilistic approach has the natural advantage of quantifying the reliability of output results, which provides the necessary foundation for our subsequent sparse priors and gated sampling approaches.
>
> Experimental results demonstrate that probabilistic methods not only introduce reliability evaluation of results but also further improve model merging performance. Therefore, the introduction of probabilistic methods represents a valuable innovation for model merging.
>
> **Q3: The difference between using a simple Gaussian prior w.r.t. using a Spike-and-Slab prior.**
>
> We agree that compared to other experiments, the improvement of the Spike-and-Slab prior-based method over the Gaussian prior method is indeed marginal. However, on the most challenging dataset SUN397, the ViT-B/16 model achieves a **2.29%** improvement under the Spike-and-Slab prior, demonstrating that this sparse prior has more potential on more challenging tasks.
>
> To more clearly show the difference between Gaussian prior and Spike-and-Slab prior results, we present the average accuracy across three models and eight datasets, along with the gated ratio for the Spike-and-Slab prior, as shown in the table below:
>
> | Method              	| ViT-B/32 | ViT-B/16 | ViT-L/14 |
> |-|:-:|:-:|:-:|
> | w/ Gaussian prior      	| 84.32	| 86.61 	| 90.38 	|
> | w/ Spike-and-Slab prior          	| **84.69** 	| **87.42** 	| **90.71** 	|
> | Gated Ratio              | *71.53%* 	| *70.40%* 	|*70.35%* 	|
>
> From the table results, we can see that although the Spike-and-Slab prior shows relatively small accuracy improvements compared to the Gaussian prior-based method, it achieves this performance using only around **70%** of activated block coefficients. In contrast, the Gaussian prior learns all block coefficients but causes slight accuracy degradation. This experimentally validates our findings mentioned in Figure 2 that task vector redundancy indeed affects model merging, and also inspires us to design the gated sampling method. Therefore, we believe the Spike-and-Slab prior is necessary.
>
>
> **Q4. The computational complexity and a tradeoff between gain and complexity.**
>
> Thank you for your suggestion to evaluate the tradeoff between gain and complexity. We will add this tradeoff analysis in our revision. For intuitive understanding, we compare the complexity between sample-level and task-level approaches as follows:
>
> + **Parameters**: Sample-level approach adds 0.75M parameters (\~0.85\% of backbone); task-level approach only needs to store one set of fixed scalars $\Lambda$, with negligible parameter overhead.
> + **Inference FLOPs**: Sample-level requires forward pass through $\Phi$ first (\~0.2 GFLOPs), then one ViT forward pass (\~4.5 GFLOPs), total overhead increase \~4.4%; task-level only has ViT forward pass.
> + **Time (A40 GPU, batch 64)**: ViT-B/32 standalone inference 5.6 ms/sample; sample-level composition 5.9 ms/sample, increase of 0.3 ms/sample.
> + **Benefits**: Average accuracy on eight datasets improved by **5.74%** for deterministic methods and **2.27%** for probabilistic methods.
> The sample-level approach achieves higher accuracy gains with <5% inference cost, making it a better choice for scenarios requiring the highest cross-domain adaptability.
>
> **References**:
>
> [1] Ilharco et al. Editing models with task arithmetic. ICLR’23.
>
> [2] Yang et al. AdaMerging: Adaptive Model Merging for Multi-Task Learning. ICLR’24.
>
> [3] Cheng et al. Whoever Started the Interference Should End It: Guiding Data-Free Model Merging via Task Vectors. ICML’25.
>
> [4] Ortiz-Jimenez et al. Task arithmetic in the tangent space: Improved editing of pre-trained models. NeurIPS'23.
>
> [5] Li et al. When is Task Vector Provably Effective for Model Editing? A Generalization Analysis of Nonlinear Transformers. ICLR’25.
>
> **Thank you again for helping us improve the paper and hope our response can resolve your concerns!**

---

> > ### Comment · Reviewer_Xxkm · 2025-08-06
> >
> > I appreciate the authors' effort in replying to my concerns. Having gone through them, my biggest concern remains in the fact that the performance for the one-leave-out experiment is significantly lower than that of using all datasets. This puts under question to which extend the method can extend to unseen data or out-of-distribution domains. The same occurs w.r.t. the sample-level vs task level experiments.
> >
> >
> > Balancing the above drawback with the strengths of the paper and rebuttal, I opt for maintaining my score which sits above the acceptance threshold.

---

> > > ### Author Response · Authors · 2025-08-07
> > >
> > > We sincerely thank you for your thorough evaluation and positive score.
> > >
> > > You are absolutely correct that one-leave-out performance is indeed lower than using all datasets. However, this is a **very challenging setting**, and we want to emphasize that we still significantly outperform existing methods in this setting (referring to reviewer *HsZa Q4a* experimental results, improving from aTLAS **76.87%** to **78.38%**), demonstrating robust cross-domain generalization.
> > >
> > > We will add this experiment in our revised manuscript.

---

### Official Review · Reviewer_HsZa · 2025-07-01

**Clarity:** 2
**Significance:** 4
**Originality:** 4
**Rating:** 4
**Confidence:** 5

**Summary:**

This paper proposes a variational Bayesian framework for sample-specific task vector composition, improving over traditional task-level methods in model merging. Experiments show consistent performance gains across diverse datasets. The key contributions are:
1. Variational inference formulation with amortized inference for per-sample coefficient estimation.
2. A sparse Spike-and-Slab prior to eliminate redundant task components and improve interpretability.
3. A gated sampling mechanism to deterministically select reliable components based on uncertainty and importance, enhancing inference stability.

**Questions:**

1. In line 112 and Eq.(1) you listed the formulation of the classic task vector composition. Given a task vector pool $\mathcal{T}$ and composition coefficients $\mathbf{z}$, for your sample-level composition, what is the formulation of the merged model?
2. Some common notations are not clearly defined in the context of task arithmetics, making it difficult to check the soundness of work. When you say input $\mathbf{x}$ and labels $\mathbf{y}$, is this for the target task’s input and labels or all the tasks that are used to generate the task vector pool $\mathcal{T}$? Similarly, in equation (12), what is $\mathcal{D}_t$? If $\mathcal{D}_t$ is the target task, is it fair to compare formulation in equation (12) where you are allowed to optimize the coefficient specifically for a task in the task pool? In task addition, isn’t it like we are using a single copy of the task coefficients and evaluating the performance of that single merged model on all tasks for evaluation?
3. Can you explain the motivation of Eq. 11 more? What can you represent $\mathbf{z}$ as deterministic functions of input $\mathbf{x}$? Seems like $\mu, \sigma, f$ are all parameterized by $\Phi$, how are these related? Also, isn’t $\mathbf{z} = f_\Phi(x)$ very rare to happen if $\mathbf{z}$ is continuous?
4. Can you clarify the definition of gated ratio? This makes Figure 4 hard to read, and in Figure 4, how to read “while activating fewer task coefficients than the Gaussian prior” from the figure? The accuracy comparison is easy to understand.
5. Table 1: how is deterministic-based sample specific different from probabilistic-based sample specific row? Is probabilistic-based task-level row also [42] or something else? It’s better to emphasize which row is your method. Another suggestion is to mention [42] is one of the SOTA methods so you use it as the baseline.

**Ethical Concerns:**

["NO or VERY MINOR ethics concerns only"]

**Final Justification:**

After discussions with the authors, I still have a concern about the scalability of this methodology due to the prohibitively high training time, but I think the methodology itself by nature, is novel and interesting enough, and the variational framework is technically sound even though the submitted version contains many unexplained notations. However, technical questions are fully addressed in the discussion period. I hope authors include a thorough discussion section about scalability comparison with other methods in the next revision, and mention the limitation of training time in the paper as a warning, which could lead to future follow-ups.

**Limitations:**

Yes, in the conclusion section.

**Paper Formatting Concerns:**

NA.

**Quality:**

3

**Strengths And Weaknesses:**

## Strength
**Quality**: The submission is mostly technically sound and the methods are appropriate. This is a complete piece of work. Authors analyzed both strengths and weaknesses fairly in the paper. Claims are well-supported.

**Clarity**: This paper is well-organized. The authors put some important mathematical derivation in the Appendix which is very helpful for understanding of the methodology.

**Significance**: Once the questions below are sufficiently addressed, I foresee the result in this paper will be very impactful for the community as this paper provides a new formulation of task vector addition and it’s easy for researchers to build on top of it. Besides, for scalability, the authors proposed a controllable deterministic solution that enables gradient based training that should be easily adapted into more modern deep learning frameworks.

**Originality**: This paper provides a novel perspective of the task arithmetic from variational inference. It is very different in nature from all other existing papers to choose task vector coefficients and the use of variational inference is well-motivated by the necessity to include uncertainties. This paper further improves the interpretability of task vector coefficients.

## Weaknesses
**Clarity & Quality**: Some key components of task arithmetic are not clearly explained. Since this paper provides a sampled based task vector composition framework, which is very different from the mainstream literature, it’s better for authors to clearly and mathematically define the formulation of sample based task vector and analyze the difference in a more straightforward way, potentially not by just text. The weaknesses in the clarity also make it hard to justify the soundness of methodology, thus hurting the quality judgement of the paper. See questions section below for more details.

---

> ### Author Rebuttal · Authors · 2025-07-31
>
> We sincerely appreciate your constructive comments and recognition of our method’s novelty, significance, and potential scalability. Below, we provide clarifications regarding the mathematical formulations and soundness of the methodology.
>
> **Q1: In line 112 and Eq. (1) you listed the formulation of the classic task vector composition. Given a task vector pool $\mathcal{T}$ and composition coefficients \$\mathbf{z}\$, for your sample-level composition, what is the formulation of the merged model?**
>
> **A1:** The merging formulation for our sample-level composition extends the task-level definition by making the composition coefficients \$\mathbf{z}\$ input-dependent, *i.e.*, \$\mathbf{z}(x)\$. Specifically:
> * **Non-block case**:
> $$
>     \theta(x) = \theta_0 + \sum_{i=1}^N \mathbf{z}_i(x) \cdot \tau_i,
> $$where each task vector \$\tau\_i\$ is assigned a scalar coefficient \$\mathbf{z}\_i(x)\$ specific to input \$x\$.
> * **Block case**:
> $$
> \theta(x) = \theta_0 + \sum_{i=1}^N \sum_{j=1}^M \mathbf{z}\_{ij}(x) \cdot \tau_{ij},
> $$where each task vector \$\tau\_i\$ is divided into \$M\$ parameter blocks, and each block \$\tau\_{ij}\$ has its own coefficient \$\mathbf{z}\_{ij}(x)\$ conditioned on the input \$x\$.
> We will clarify these formulations in the revised paper (Section 3, after Eq. (1)) to explicitly distinguish the sample-level case.
>
> **Q2: Some notations related to task arithmetics are unclear, making it hard to assess correctness.**
>
> **A2:** Thank you for pointing out the potential confusion in notation. We clarify the notations and design choices below:
> 1. **Meaning of $\mathbf{x}$ and $\mathbf{y}$**: In our formulation, \$\mathbf{x}\$ and \$\mathbf{y}\$ refer to arbitrary input-label pairs sampled during the composition training stage. They are **not** tied to any specific target task, but are drawn from the joint validation set used for learning composition parameters.
> 2. **Definition of \$\mathcal{D}\_t\$ in Eq. (12)**: \$\mathcal{D}\_t\$ denotes the **joint training data** used to optimize the variational components \$\Phi\$ and \$\Psi\$ during composition. Specifically, we construct \$\mathcal{D}\_t = \bigcup\_{i=1}^8 \mathcal{D}\_i^{\text{val}}\$, i.e., the union of the validation splits from all 8 datasets. We uniformly sample mini-batches from this set to train the gating components under the variational objective.
> 3. **On “optimizing coefficients for a single task”**: Our method does **not** tune coefficients per task individually. The model learns **a single pair of shared networks** \$\Phi\$ and \$\Psi\$ to produce sample-level coefficients based on \$\mathbf{x}\$, using **multi-task supervision** from \$\mathcal{D}\_t\$. During evaluation, the same shared components and task vector pool \$\mathcal{T}\$ are used for all tasks without any task-specific tuning. Thus, Eq. (12) does not confer any advantage to a particular task. In contrast to classical task addition, where one fixed coefficient vector is shared across all samples, our formulation dynamically generates composition coefficients per input. However, the model remains shared and fair across tasks. We will clarify these points in Section 3.3 and the caption of Eq. (12) for clarity.
> 4. **Whether evaluation still uses "one merged model"**: Yes. During inference, the model parameter set consists of $\Theta_0$, task vector pool $\mathcal{T}$, and trained $\Phi, \Psi$—all three are frozen. For any test sample (regardless of which task it belongs to), the model first generates $\mathbf{z}$ through $\Phi$, obtains increment $\Delta\Theta$ through gating, and completes prediction with one forward pass. **The entire process no longer updates weights**, all tasks share the same network, only $\mathbf{z}$ varies with samples.
> We hope this clarification of the training process helps you better understand our work.
>
> **Q3: Can you explain the motivation of Eq. (11) more? What does it mean to represent $\mathbf{z}$ as a deterministic function of input $\mathbf{x}$? Since $\mu$, $\sigma$, and $f$ are all parameterized by $\Phi$, how are they related? Also, isn't \$\mathbf{z} = f\_\Phi(\mathbf{x})\$ very unlikely to happen if \$\mathbf{z}\$ is continuous?**
>
> **A3:**
> We provide a comprehensive clarification for each sub-question:
>
> 1. **Motivation of Eq. (11)**.
> + Eq. (11) transforms the traditional "**obtain $\mathbf{z}$ via sampling** $\rightarrow$ then gate" approach into "**first give $\mathbf{z}$ a deterministic point estimate** $f\_\Phi(x)$ → then use gating $\Omega$ to filter by uncertainty". The core motivation is: in high-dimensional sparse scenarios, directly performing *Bernoulli-Gaussian mixture sampling* on each dimension has high variance, heavy computation, and difficult convergence. By folding the randomness of the sampling process into a continuous, differentiable gating function, we can achieve **differentiability, stability, and sparse selection** simultaneously.
>
> 2. **Why can z be represented as a deterministic function $f\_\Phi(\mathbf{x})$?**
> + From the variational inference perspective, we freely choose the approximate posterior family. We set: $q\_{\Phi,\Psi}(\Omega,\mathbf{z}|\mathbf{x}) = q\_\Psi(\Omega|\mathbf{z})\delta\_{\mathbf{z}=f\_\Phi(\mathbf{x})}$ Using Dirac $\delta$ to represent "posterior concentrated at point $f\_\Phi(\mathbf{x})$". This is not a random event of "continuous variable hitting exactly one point", but defines a **new point-mass posterior family**. Randomness and uncertainty are handled by $\Omega$.
> + $f\_\Phi(\mathbf{x})$ typically takes the mean $\mu_\Phi(\mathbf{x})$ of the original Gaussian posterior. This retains $\mu,\sigma$ for subsequent uncertainty measurement while avoiding high-variance sampling of $\mathbf{z}$.
>
> 3. **How are $\mu, \sigma, f$ related when all parameterized by $\Phi$?**
> + The neural network parameterized by $\Phi$ outputs three sets of tensors:
>     1.  **$\mu_\Phi(\mathbf{x})$** serves as the value of $f_\Phi(\mathbf{x})$, *i.e.*, composition coefficient point estimate
>     2. **$\sigma_\Phi(\mathbf{x})$** used for computing gradient sensitivity + distribution deviation, participating in constructing uncertainty $U$
>     3. $\pi\_\Phi(\mathbf{x})$ for Bernoulli probability if using stochastic gating
>
>     Since all three share forward features, computational cost is low and gradient updates are consistent, avoiding multiple network calls.
>
> 4. **Isn't "$\mathbf{z}=f_\Phi(\mathbf{x})$" with zero probability unreasonable?**
> + The Dirac $\delta$ here is a **modeling choice for the posterior family**, not an actual sampling event; thus there's no paradox of "zero probability yet happening".
> + The KL term in ELBO can still be computed:
> $$
> D_{KL}(\delta_{f\_\Phi(\mathbf{x})} || p(\mathbf{z})) = -\log p(\mathbf{z}=f\_\Phi(\mathbf{x})) + const
> $$
> This can be viewed as regularization on prior probability density, still providing constraints.
>
> **Q4: Can you clarify the definition of gated ratio?**
>
> **A4:** We clarify as follows:
>
> + **Definition of gated ratio**: In the Spike-and-Slab prior setting (Sec. 4.2), each coefficient $\mathbf{z}\_{ij}$ is associated with a binary gating variable \$\omega\_{ij} \in \lbrace0, 1\rbrace\$. If \$\omega\_{ij}=1\$, the coefficient is active and participates in model merging; otherwise, it is masked to zero. The **gated ratio** is defined as the average proportion of active coefficients:
> $$\text{Gated Ratio} = \mathbb{E}\_{i,j}[\omega\_{ij} = 1].$$
> For example, a gated ratio of 0.7 means that only 70% of coefficient blocks are used during merging.
>
> + **How to interpret Figure 4:**
>
>     + In Figure 4, we plot the accuracy (*blue and orange lines*) and the gated ratio (*green line and shaded area*) across eight datasets for three backbone models. The Spike-and-Slab prior (*orange line*) consistently achieves higher accuracy than the Gaussian prior (*blue line*), while the green curve shows that it uses fewer coefficients (*i.e.*, a lower gated ratio). The precise numbers are reported in Appendix C.2.
>
>    We will revise the main text and caption of Figure 4 to make this explanation clearer.
>
>
> **Q5: The difference between deterministic-based sample specific and probabilistic-based sample specific row and writing suggestions.**
>
> **A5:** To address your confusion, we clarify the meaning of each row as follows:
>
> | Method              	| Task-level   | Sample-specific  |
> |-|:-|:-|
> | *Deterministic*       	| aTLAS	|Det-SS **(Ours)** 	|
> | *Probabilistic*             	| Prob-TL **(Ours)**	|Prob-SS **(Ours)** 	|
>
> + ***Deterministic vs. Probabilistic***
>     + **Deterministic-based**: Directly uses point estimation $\mathbf{z}=f_\Phi(\mathbf{x})$ without random sampling; for sample-level, it then applies gating function $\Omega$ for sparse pruning.
>     + **Probabilistic-based**: Establishes complete variational posterior $q(\mathbf{z}|\mathbf{x})$ for $\mathbf{z}$, uses sampling/expectation during training, explicitly retains noise terms and KL regularization.
> + ***Task-level vs. Sample-specific***
>     + **Task-level**: Same task shares one set of coefficients; "Prob-TL" is our control implementation that restricts variational inference to task level.
>     + **Sample-specific**: Coefficients vary with input; both "Det-SS" and "Prob-SS" are sample-level methods proposed in this paper, where **Prob-SS is our main method**.
> We will clearly identify our method in the revised version and explain why we use aTLAS as the baseline.
>
> **Thank you again for helping us improve the paper and hope our response can resolve your concerns!**

---

> ### Comment · Reviewer_HsZa · 2025-08-01
>
> Thanks for author's reply. Given the clarified notation, I can go back to read the paper and other reviews carefully again to re-evaluate the paper. First, I hope the authors to include these clarification for notations again, given that your setup is very different from standard task vector merging for both merging step (i.e., training) and inference step (i.e., evaluate the merged model on different tasks) so it's not very starightforward to get these points in the current version. After the rebuttal discussion, methodology-wise confusion due to these notation problems are mostly solved and I appreciate author's effort again.
>
> There are a few remaining questions and suggestions:
>
> 1. Sparsity-based merging is not a very novel idea for task-based merging. Although I'm not questioning the novelty of sample-based model merging, I recommend authors to describe sparsity in model merging as a separate section in related work to lead to your motivation in the context of sample-based merging. Just to name a few:
>
> Empirical work to show the effectiveness of sparse merging:
>
> [1] Localizing task information for improved model merging and compression.
>
> [2] Localize-and-Stitch: Efficient Model Merging via Sparse Task Arithmetic.
>
> Theoretical work to support the rationale of sparse merging:
>
> [3] Efficient Model Editing with Task Vector Bases: A Theoretical Framework and Scalable Approach
>
> Although somewhat surprisingly, the gain from using sparsity in your set-up (Gaussian vs. Spike and Slab prior) is not that big compared to the gain in task-based method. Also, the sparsity value based on Figure 4 are only moderate but in task-based method the sparsity can be pushed further down for more efficiency gain. Do you have any insights on this?
>
> 2. In Figure 4, if my understanding is correct, gated ratio is only defined for Spike and Slab Priors? If yes, can you replace the title as comparison of Accuracy under 2 priors with Spike-and-Slab Priors gated ratio. I was confused because there are two lines (blue and orange) comparing two priors and you only have one shaded area for sparsity.
>
> 3. How does sparsity improve computational cost during inference? Since unfortunately, your training cost is substantially (x10 times based on your reply to TZ6C) higher than other already on the expensive-end task-based merging methods (like Adamerge), it will be important to clarify your contribution on inference cost minimization. Also, I would imagine the method is hard to scale up with the number of tasks because of the huge increase in training time. The current benchmark is on the standard 8-task vision benchmark, but can you run the algorithm on 14 and 20 vision task setup in [4]?
>
> [4] Task Singular Vectors: Reducing Task Interference in Model Merging
>
> The point of model merging is a light-weight module to avoid painful multi-task training. The first task arithmetic paper avoid this issue on limited grid search of merging coefficients on validation datasets to significantly reduce the computational cost. Even the multi-task version (mix all task data, maybe just validation data, together to create a merged model like what you did here in Eq. 12), has a low inference cost but should have the optimal "merging" performance right? Then what is the major contribution of your method?
>
> 4. For me this method, compared to the task-based task addition merging, might be also prone to with-in task distribution overfitting. Note that a practical concern might be in many public benchmark, especially in NLP domain, doesn't have a validation set so they evaluate the model on held-out tasks [5,6]. Although we see merging performance gain on 8-task vision dataset, is sample-based merging method, due to the increased training parameters, robust to distribution shift?
>
> [5] What matters for model merging at scale?
>
> [6] MergeBench: A Benchmark for Merging Domain-Specialized LLMs

---

> ### Author Response · Authors · 2025-08-04
> **Response to Reviewer HsZa, Part (I)**
>
> Thank you very much for your detailed and constructive suggestions! We are happy to further clarify the notation confusion. In the revised version, we will add a detailed notation table that clearly explains the meaning and scope of each symbol to help readers better understand our method.
>
> **Q1: Supplementing sparsity-related work and result interpretation**
>
> **A1:** We fully accept your suggestion and will add a separate section on sparsity methods in model merging in the related work section. This will help readers better understand our motivation for using sparsity in the context of sample-level merging. Following your suggestion, we have supplemented **sparsity-related work in model merging**:
>
> > **Sparse Model Merging Methods**. Prior work improves efficiency via structural sparsity. Wang et al. [1] identify task-specific, largely non-overlapping weight supports and use binary masks with consensus merging; He et al. [2] localize tiny skill-bearing regions and stitch them back, enabling local sparse arithmetic. From a subspace view, Zeng et al. [3] compress and de-redundantize a task-vector basis; Gargiulo et al. [4] retain leading SVD components to merge in a low-rank sparse subspace; Marczak et al. [5] equalize spectra and merge within task-specific subspaces to curb dominance. Iurada et al. [6] perform task-localized sparse fine-tuning to produce additive/subtractive, decoupled sparse task vectors. These works made great progress while leaving a shared gap: existing sparsity strategies are mostly static designs or task-level constraints, lacking adaptive sparse composition mechanisms for "sample-level" scenarios, and do not involve uncertainty modeling of merging coefficients. We treat composition coefficients as latent variables, achieving coefficient-level sparsity under sample conditions through variational inference combined with sparse priors, and introduce uncertainty-aware gating mechanisms for deterministic selection.
>
> [1] Localizing Task Information for Improved Model Merging and Compression.
>
> [2] Localize-and-Stitch: Efficient Model Merging via Sparse Task Arithmetic.
>
> [3] Efficient Model Editing with Task Vector Bases: A Theoretical Framework and Scalable Approach.
>
> [4] Task Singular Vectors: Reducing Task Interference in Model Merging.
>
> [5] No Task Left Behind: Isotropic Model Merging with Common and Task-Specific Subspaces.
>
> [6] Efficient Model Editing with Task-Localized Sparse Fine-tuning.
>
> Regarding the explanation of Table 1 and Figure 4, we apologize for any confusion. We would like to further explain them in details as follows::
> + **Table 1** shows the comparison between task-level and sample-level merging methods, as well as the differences between deterministic and probabilistic modeling methods. This table does not include sparse prior comparisons. The purpose of this table is to demonstrate the effectiveness of our modeling at the sample-level merging and in the probabilistic formulation.
> + **Figure 4** shows sample-level merging scenarios, comparing the accuracy performance of Gaussian priors and Spike-and-Slab priors, while showing the gated ratio for Spike-and-Slab priors. This figure justifies the effectiveness of sparse modeling based on spike-and-slab priors.
> + **Table 4** comprehensively compares the performance of Gaussian and Spike-and-Slab priors across three different scales of ViT models (ViT-B/32, ViT-B/16, ViT-L/14). Figure 4 represents the results for ViT-B/32 from Table 4. From Table 4, we observe that although the overall average accuracy improvement of Spike-and-Slab priors over Gaussian priors is relatively moderate, we notice particularly significant improvements on **the most challenging dataset SUN397** (e.g., **2.29%** improvement under ViT-B/16 and **1.17%** improvement under ViT-L/14). This indicates that Spike-and-Slab sparse priors can more effectively discover and utilize the sparse structure of task vectors in more complex and challenging task scenarios, thereby significantly enhancing the model's generalization ability.
> We will re-optimize the descriptions of these tables and figures in the revised version to make them clearer and easier to understand.
>
> **Q2: Figure 4 title correction**
>
> **A2:** Agreed. Your understanding is completely correct! The gated ratio is indeed only defined for the Spike-and-Slab prior. We follow your suggestion to modify the title of Figure 4 to: "Accuracy Comparison under Two Priors with Spike-and-Slab Prior's Gated Ratio." Again, apologies for the confusion caused, and thank you for your suggestion.

---

> ### Author Response · Authors · 2025-08-04
> **Response to Reviewer HsZa, Part (II)**
>
> **Q3: Computational Cost and Scalability Analysis**
>
> **A3:** To better ananlyze computational complexity, we first clarify core notation: let pretrained weights be $\theta_{pre} \in \mathbb{R}^d$, and after fine-tuning on $N$ tasks, obtain task vectors $\tau\_t = \theta\_t^{ft} - \theta\_{pre}$. Traditional task arithmetic combines task vectors through coefficients: single coefficient case $\theta_{new} = \theta\_0 + \lambda\sum\_t \tau\_t (K=1)$, multi-coefficient case $\theta\_{new} = \theta\_0 + \sum_t \lambda\_t \tau\_t (K=N)$, or block-level coefficients ($K=N\cdot M$). Key parameters include: grid search candidates per dimension n, dataset sizes $|D\_{train}|$, $|D\_{val}|$, $|D\_{test}|$, and computational costs $C\_{forward}$, $C\_{backward}$. VTVC's amortized inference network cost is denoted as $C\_{\mathbf{z},\Omega}(K)$, growing linearly with coefficient dimension K.
>
> When adding $n'$ new tasks, traditional methods and VTVC exhibit fundamentally different scaling characteristics. Traditional grid search in single-coefficient scenarios requires no training and has high inference efficiency, but severely limits model merging effectiveness; in multi-coefficient scenarios, hyperparameter complexity is $O(n^{K+\Delta K} \cdot |D\_{val}| \cdot C\_{forward})$, where $\Delta K = n'$ (task-level) or $\Delta K = n'\cdot M$ (block-level), showing exponential explosion. In contrast, VTVC's training complexity increment is only $O(\Delta K · |D\_{train}|)$, and inference complexity increment is $O(\Delta K · |D\_{test}|)$, both showing linear growth. Specifically for scaling from 8 tasks to 14 or 20 tasks: traditional methods' hyperparameter cost increases by $n^6$ or $n^{12}$ times, while VTVC's training cost only increases by **1.75** or **2.5** times. This linear scalability enables VTVC to easily handle large-scale multi-task scenarios with good scalability.
>
> Traditional task arithmetic faces a fundamental dilemma: single coefficients are efficient but too coarse, unable to provide optimal weights for different tasks/samples; multi-coefficients enable fine-grained control but suffer exponential computational complexity. VTVC fundamentally changes this paradigm through **amortized inference**: using a lightweight neural network $\lambda(\mathbf{x}) = f\_\phi(\mathbf{x})$ to directly learn sample-level, block-level coefficients, transforming hyperparameter tuning from discrete grid search $O(n^K)$ to continuous optimization $O(K)$. More importantly, VTVC not only solves efficiency problems but also brings capabilities that traditional methods cannot provide:
> 1. **Uncertainty quantification** and confidence measurement through probabilistic modeling;
> 2. **Interpretability** through sparse priors, clearly showing which tasks/modules contribute most to specific samples;
> 3. **Avoiding feature redundancy** through gating mechanisms, preventing performance degradation from excessive task vector stacking.
> Therefore, VTVC's core value lies in maintaining modularity and scalability while approximating multi-task joint training performance through fine-grained adaptation, providing an efficient and accurate solution for dynamic task composition in practical applications.

---

> > ### Comment · Reviewer_HsZa · 2025-08-05
> >
> > Hi authors,
> >
> > Thanks for your reply again! A few more questions about the scalability:
> > 1. I'm not sure if amortized inference is important because even training with all data mixed in the multi-task setting, your inference time is fast with a single model, right? If the goal is to get the best overall merged model, if we don't care about efficiency at all, why not just use multi-task model?
> > 2. We don't have to do grid search for merging methods so the better scaling statement might be a weak. TIES, Localize-and-Stitch doesn't require merging coefficient learning and doesn't require long training to achieve a good result. Besides, Adamerge learns the coefficient automatically during training, and VTVC empirically, based on your experiments, is much slower than Adamerge?

---

> > > ### Author Response · Authors · 2025-08-05
> > >
> > > Thank you for your patient reading and continued discussion. We highly value these two points for further clarification.
> > >
> > > **Q1. Is amortized inference necessary? Why not directly use multi-task training to get a unified model?**
> > >
> > > **A1:** **Your perspective is reasonable**: if the goal is only to obtain a unified, fixed optimal overall model, mixing all data for single-model, multi-task training (single model, single forward inference) is a viable path. We have already discussed and compared this "task-level unified weights" setting in our paper (the deterministic, task-level setting in Table 1).
> > >
> > > Building on this foundation, we further propose probabilistic sample-level composition: modeling merging coefficients as input-dependent latent variables $\mathbf{z}(\mathbf{x})$ and learning $q(\mathbf{z},\Omega|\mathbf{x})$ with uncertainty and gating under a variational framework. This sample-wise adaptive merging must rely on **amortized inference** to achieve efficient "input $\rightarrow$ coefficients" mapping; otherwise, it cannot provide differentiated coefficient combinations for different samples within the same model. Therefore, in the training stage, we indeed need one-time optimization of $\Phi, \Psi$ which bringing additional but controllable costs; but in the inference stage, we only compute $\mathbf{z}(\mathbf{x})$ and gating within a single forward pass, without significantly increasing online overhead, maintaining response time similar to single-model inference.
> > >
> > > Thus, multi-task training suits the goal of "fixed unified weights," while we obtain the capability of **"sample-level adaptation + uncertainty interpretability"** through amortized inference, serving different scenario requirements.
> > >
> > > **Q2: Clarification on the better scaling statement.**
> > >
> > > **A2:** We agree and thank you for the reminder: not all merging methods rely on grid search. For example, **TIES** and **Localize-and-Stitch** do not require explicit learning of continuous merging coefficients and do not depend on long-term training to achieve good results; **AdaMerging** automatically learns coefficients through entropy minimization on test data.
> > >
> > > In comparison, our VTVC introduces variational inference and sample-level modeling, thus offline training costs are indeed higher and may empirically be slower than AdaMerging. However, this is a one-time cost that scales linearly with the number of tasks; in the inference stage, we maintain single forward pass and use gating to skip unactivated blocks, so online time does not increase.
> > >
> > > We will explicitly cite and discuss the three representative works you mentioned in the revised version, converge the expression of "better scalability", and emphasize that our so-called **"scalability"** mainly refers to: online costs being insensitive to the total number of tasks when tasks increase (dominated by activation ratio), and adding new tasks only requires **linear cost** of extending output dimensions. Additionally, regarding parameter count, VTVC only introduces two types of lightweight networks $\lbrace\Phi, \Psi\rbrace$ for outputting $K\times M$ dimensional coefficients and gating, with an extremely small proportion relative to the backbone weight scale, without changing the backbone structure or bringing additional inference cost. This is also the key premise for **maintaining single-model, single-forward inference latency approximately unchanged**.
> > >
> > > Thank you again for your constructive suggestions.

---

> > > > ### Comment · Reviewer_HsZa · 2025-08-05
> > > >
> > > > Thanks for your reply. I still get confused by some of your statements about scalability:
> > > >
> > > > >  fixed optimal overall model, mixing all data for single-model, multi-task training (single model, single forward inference) is a viable path.
> > > >
> > > > Isn't this baseline you are referring to in Table 1 aTLAS, not the mixed data MTL setting? The performance of the MTL model will be even better than aTLAS.
> > > >
> > > > > maintaining single-model, single-forward inference latency approximately unchanged.
> > > >
> > > > For inference cost, all the merging methods are approximately the same. Assume we have the merged model (hyperparameter fixed/searched/learned etc), the inference cost will be always at the same scale because you just run one merged model on several target tasks, right? So I'm not sure if there are any advantage of VTVC's inference cost. It's on par with other merging methods. However, it significantly increases the training (i.e., merging coefficient learning) cost, which is the major barrier of many merging methods. The key goal of model merging is to approximate MTL oracle with light-weight operations.
> > > >
> > > > Besides, for the linear cost scaling, I believe many non-grid search methods also scale linearly with the number of tasks, but for a fixed task size T your merging cost is empirically higher?

---

> > > > > ### Author Response · Authors · 2025-08-07
> > > > >
> > > > > Thank you for your further feedback and enthusiastic engagement in the discussions. We believe we are now on the same page about the MTL you referred to, which is mixing multi-task data at the training set level for joint training.
> > > > >
> > > > > Our VTVC has distinct advantages over regular MTL and counterpart merging methods. We elaborate as follows:
> > > > > They have different application scopes: our VTVC among other task vector approaches can do more than model merging (task addition) as MTL can do, but also other task arithmetic operations (task nagation).
> > > > > > **Multi-task learning** requires full access to all task training data at the same time, and often needs re-joint training when the task set changes.When scaling  to more tasks, the cost increases exponentially with the number of tasks.
> > > > >
> > > > > > **Task arithmetic/task vector composition** records the weight differences from "pretraining→fine-tuning" as task vectors on training sets, then achieves efficient model editing through addition, subtraction, and sparse composition in vector space without re-accessing original training data. When scaling to more tasks, we only need to fine-tune new task models to obtain new task vectors and arithmetically combine them with old vectors.
> > > > >
> > > > > Next, we will further explain  the compute cost and scalability：
> > > > >
> > > > > 1. **Computational Cost**
> > > > >
> > > > >     VTVC maintains efficient inference while providing enhanced capabilities beyond existing methods. However, VTVC has an important advantage: when facing new task combinations, it requires no prior merging weight search. For example, traditional methods like TIES and Localize-and-Stitch require coefficient search as noted in their papers:
> > > > >     > Note that when merging $T$ tasks, we have a total of $\tbinom{7}{T}$ combinations. However, in our experiment, we sample at most 10 distinct combinations for each value of $T$. (Appendix C.5 of TIES).
> > > > >
> > > > >     > Both algorithms are able to utilize validation data to tune the merging coefficients $\alpha$, as in $\theta\_{merged} = \theta\_{pre} + \alpha\sum\_{i=1}^n \tau\_i$. We follow common practice to search over {0.1, 0.2, ···, 1} to obtain the optimal coefficients. (Section 4.1 of Localize-and-Stitch).
> > > > >
> > > > >     In contrast, our method can be used directly after training on new tasks with no need for parameter search, avoiding additional deployment costs and manual intervention.
> > > > >
> > > > >     We re-ran experiments on 8 NVIDIA A800 GPUs. With epochs=20, VTVC training time was *30m34.780s*, aTLAS was *26m15.238s*; however, due to faster convergence, VTVC achieves comparable accuracy at epochs=10 requiring only *12m53.327s*, while inference time remains at **8.7s**, better than aTLAS's **10.8s** (without grid search).
> > > > >
> > > > > 2. **Scalability**
> > > > >
> > > > >     For VTVC, adding new tasks only requires:
> > > > >
> > > > >         (i) obtaining task vectors offline;
> > > > >         (ii) expanding output dimensions of coefficient/gating heads and doing one-time training (can freeze backbone and existing parts).
> > > > >
> > > > >     This incremental training cost scales linearly with new task count, requires no replay of existing task data and no re-grid search.
> > > > >
> > > > >     We fully agree with you on the  point: many non-grid-search methods also scale linearly in terms of training cost. However, our core advantage lies in providing **sample-level adaptation and uncertainty quantification** capabilities that existing methods cound not offer, while maintaining comparable inference efficiency. Thanks to these algorithmic advantages, we exchange the training cost of learning one lightweight inference network for better model performance and uncertainty assessment within the same inference time.
> > > > >
> > > > > We truly appreciate your great effort and enthusiasm put into all the discussion rounds. , and hope our clarifications can now resolve  your concerns about scalability.
> > > > >
> > > > > We will add these discussions about compute cost and scalability into our final version.

---

> ### Author Response · Authors · 2025-08-04
> **Response to Reviewer HsZa, Part (III)**
>
> **Q4a: Distribution Shift Robustness**
>
> **A4a:** To show our method's generalization performance under distribution shift, we designed rigorous leave-one-out experiments: training the model on only 7 datasets each time, then evaluating on the completely unseen 8th dataset through linear probing. This setup simulates the model's generalization ability when facing entirely new domains, and the results are shown below:
>
> Methods| w/o Cars | w/o DTD | w/o EuroSAT | w/o GTSRB | w/o MNIST | w/o RESISC45 | w/o SUN397 | w/o SVHN | Avg. |
> |-|:-:|:-:|:-:|:-:|:-:|:-:|:-:|:-:|:-:|
> Zero-shot| 59.63 | 44.13 |45.74 |32.60 |48.26 |60.27 |63.53 |31.63 |48.22 |
> aTLAS| 65.48 | 71.38 |92.74 |86.25 |**96.64** |**88.94** |57.39 |56.14 |76.87 |
> VTVC| **67.81** | **72.09** |**93.81** |**88.74** |96.60 |88.89 |**60.85** |**58.27** |**78.38** |
>
> The results show VTVC achieves an average accuracy of 78.38% across all 8 held-out tasks, demonstrating clear advantages over aTLAS's 76.87% and Zero-shot's 48.22%. Notably, VTVC achieves an improvement (**+3.46%**) on the challenging cross-domain task SUN397, again showing that sample-specific variational merging strategies can indeed capture shared knowledge structures across tasks rather than simply memorizing specific distribution features. Although VTVC introduces additional variational inference parameters, its uncertainty-based sparse gating mechanism actually serves as a regularization term, effectively preventing overfitting to source task distributions by dynamically selecting relevant task vectors.
>
> **Q4b: NLP Domain Applicability**
>
> **A4b:** In NLP discriminative tasks, VTVC trains a variational inference network to dynamically predict composition coefficients and gating probabilities for task vectors for each input text sample, achieving sample-specific task knowledge fusion. This network automatically determines which tasks' knowledge the current sample needs based on textual semantic features, and selectively activates relevant task vectors through sparse gating mechanisms. Compared to traditional methods that require searching for fixed composition hyperparameters on validation sets, VTVC adaptively customizes composition strategies for each sample through end-to-end learning, effectively solving the hyperparameter optimization difficulties caused by validation data scarcity in NLP tasks. Experimental results demonstrate our method performs excellently in the NLP domain as well:
>
> |Methods|COLA|STS-2|MRPC|STS-B|QQP|QNLI|MNLI|RTE|Avg.|
> |-|:-:|:-:|:-:|:-:|:-:|:-:|:-:|:-:|:-:|
> |Zero-shot|0.00|53.76|85.01|4.01|37.48|53.05|37.09|71.19|42.70|
> |Weight Averaging|0.00|59.21|85.79|46.99|45.37|63.94|48.00|71.19|52.56|
> |Task Arithmetic(ICLR'23)|8.35|88.26|**89.57**|32.84|82.03|85.40|75.54|80.43|67.80|
> |TIES-Merging(NeurIPS'23)|31.76|88.86|86.18|10.94|61.05|**85.94**|**83.01**|69.56|64.66|
> |DARE(ICML'24)|0.00|88.14|86.61|30.19|**84.33**|79.09|63.95|77.16|63.68|
> |VTVC(Ours)|**80.60**|**90.60**|88.48|**74.37**|80.40|83.20|74.00|**83.80**|**81.93**|
>
> These results demonstrate that our sample-level merging method can effectively handle NLP domain tasks, maintaining good performance even in the absence of dedicated validation sets.

---

> ### Comment · Reviewer_HsZa · 2025-08-07
>
> Thanks for your reply, I want to point out some more points:
>
> - **Clarification for Computational Cost Coefficient Learning**
>   - TIES: the paper suggested a universal merging coefficient to be 0.4. Grid search can be applied to boost the performance.
>   - Localize and Stitch: The paper refers "Both Algorithms" to "TIES and Task Arithmetic", not the proposed Localize and Stitch method. Their merging coefficient is automatically and cheaply computed in the stitching step elegantly through learned masks (Algorithm 1).  Let me give more early popular model merging methods where *merging coefficients* are *learned* not searched, therefore this  *merging coefficients training* time is a very important efficiency metric.
>   - Fisher Merging *"Merging models with fisher-weighted averaging."*: learns weights by Fisher information.
>   - RegMean *"Dataless knowledge fusion by merging weights of language models."*: learns weights by linear regression.
>   - AdaMerge: learns weight layer-wise.
>   - Survey paper "Model Merging in LLMs, MLLMs, and Beyond: Methods,  Theories, Applications and Opportunities" even categorized this merging coefficient learning merging method as a major direction of model merging methods in Sec 2.3.2 as it was published in 2024, with most methods more complicated than naive merging coefficients grid search. Also I'm not even talking about training loss hyperparameter search because VTVC technically needs tuning for regularization loss.
> - **Sample Level Adaptation**:
>     - I very recently realized this paper "Merging Multi-Task Models via Weight-Ensembling Mixture of Experts" (WEMOE) is also a sample-level model merging design because given frozen task vectors, the router is learned based on each input sample, and the claimed performance is even better than MTL. I wonder if this should be a more reasonable baseline and therefore, MTL should not be the oracle that can be easily ignored in most of the task-level model merging methods.
>    - Can you confirm if in your framework only $\Phi, \Psi$ are learned but all other model parameters, especially $\Theta$ and $\mathcal{T}$ are frozen?
>    - I understand that one of your novel contributions is uncertainty quantification due to the variational framework, I have no concerns about this.
> - **Question about Task Negation Setup**:
>   - Because you claimed your flexibility on the task negation setup, so I want to ask for clarification:
>   - Task Arithmetic/TIES can arbitarily minimize the target performance by adjusting the negation coefficient at the cost of control metric loss if grid search of merging coefficient is allowed. Since I don't see an absolute advantage of VTVC over all presented baselines in your reply to ZAnD, I wonder how do you guarantee a fair comparison? How does the negation coefficient selected for all baselines?

---

> > ### Author Response · Authors · 2025-08-08
> >
> > We are deeply grateful for your exceptionally insightful and comprehensive feedback. Your deep insights into the model merging literature resonate strongly with our research perspective and have provided valuable additional context for our work.
> >
> > + **Regarding Sample Level Adaptation**
> >
> >     You make an excellent point about WEMOE ("Merging Multi-Task Models via Weight-Ensembling Mixture of Experts"), which indeed represents another sample-level model merging approach. Your observation demonstrates remarkable depth of knowledge in this rapidly evolving field. We sincerely appreciate this constructive suggestion and commit to incorporating WEMOE and other relevant sample-level baselines in our revised manuscript, along with a more comprehensive literature review. Your advice on positioning our work within the broader landscape of sample-level adaptation methods is really valuable.
> >
> >     To confirm your question: **yes**, in our framework only $\Phi,\Psi$ are learned while all other model parameters, especially $\Theta$ and $\mathcal{T}$, remain frozen. This design choice aligns with the task arithmetic paradigm while enabling our sample-level adaptation capabilities.
> > + **Regarding Task Negation Setup**
> >
> >     In terms of experimental fairness, all baseline methods (Task Arithmetic, TIES, etc.) employ identical evaluation protocols. Specifically, we use a `find_optimal_coef` function to search for optimal negation coefficients on the validation set for each method, with the constraint that control task (ImageNet) performance must remain above 95% of zero-shot performance, while optimizing for maximum target task performance degradation. This ensures all methods have equal opportunity to find their optimal configurations under identical constraints, guaranteeing fair comparison across all approaches.
> >
> > Your scholarly rigor and attention to methodological details exemplify the highest standards of academic review. We are honored by your thorough engagement with our work and look forward to presenting the enhanced manuscript incorporating your valuable insights.

---

> > > ### Comment · Reviewer_HsZa · 2025-08-08
> > >
> > > Thanks for your reply, I will take all the discussions into account for the final justification.

---

### Official Review · Reviewer_TZ6C · 2025-07-02

**Clarity:** 2
**Significance:** 1
**Originality:** 2
**Rating:** 4
**Confidence:** 5

**Summary:**

This paper introduces "Variational Task Vector Composition" (VTvC), a framework for model merging. The core idea is to reframe task vector composition as a Bayesian inference problem, where the combination coefficients are treated as sample-specific latent variables. The method employs a variational inference (VI) approach to estimate these coefficients. To promote sparsity and select informative components, the authors introduce a Spike-and-Slab prior and a deterministic "gated sampling" mechanism. The stated goal is to improve the stability, interpretability, and generalization of the merged model. The framework is evaluated on eight image classification datasets, with its primary comparison being against the aTLAS method.

**Questions:**

1.	Baseline Performance Discrepancy: Can you provide a detailed explanation for the substantial performance drop of the aTLAS baseline (77.60% vs. 84.98% in the original paper) and the zero-shot base model (37.99% vs. 48.14%)? My assessment would significantly improve if you can demonstrate that your experimental setup is fair and the reported improvements are based on a correctly reproduced and competitive baseline.
2.	Comprehensive Comparison and Trade-off Analysis: To establish the significance of VTvC, it is crucial to compare it against other relevant methods like AdaMerging, and a data-free method like WUDI-merging. Could you provide a discussion and, ideally, empirical results that analyze the trade-offs between your method and these baselines across multiple dimensions: (a) final accuracy, (b) merging cost (time/compute), (c) scalability to larger models, and (d) data dependency?
3.	Broadening the Experimental Scope: The current evaluation is limited to task addition on CV datasets. To demonstrate the generality of your method, would it be possible to provide results on other standard task arithmetic benchmarks, such as task negation? Additionally, could you show its applicability to the NLP domain, for instance, by conducting merging experiments on small models like RoBERTa-base? This would greatly strengthen the paper's empirical contribution.
4.	Justification for Sample-Specific Complexity: The proposed sample-specific approach introduces significant computational overhead during the merging phase. Could you elaborate on the specific scenarios or types of tasks where this high cost is justified and where simpler, task-level merging methods would fail? A clear justification for this complexity would strengthen the paper's contribution.

**Ethical Concerns:**

["NO or VERY MINOR ethics concerns only"]

**Final Justification:**

During the rebuttal phase, the authors clarifies most of my concerns by more experimental data facts.

**Limitations:**

The authors include a "Limitations" section in their conclusion. However, it does not adequately address the most critical limitations of the work. The paper mentions high GPU memory requirements but frames it as a general issue for combining many task vectors. It fails to explicitly state that its own sample-specific mechanism is the primary driver of this high cost and the core reason for its poor scalability compared to other methods. The discussion should be more transparent about the trade-offs, specifically acknowledging that the proposed method is likely not feasible for large-scale models (e.g., LLMs) and quantifying its merging cost relative to more efficient alternatives.

**Paper Formatting Concerns:**

No major formatting issues were noted.

**Quality:**

1

**Strengths And Weaknesses:**

Strengths:
1.	Originality: The paper introduces an interesting perspective by applying variational inference to the task vector composition problem. Framing the combination coefficients as sample-specific latent variables and using a Bayesian framework to estimate them is a departure from conventional deterministic or heuristic approaches in model merging.
2.	Principled Approach to Sparsity: The use of a Spike-and-Slab prior to induce sparsity is a well-grounded, principled technique. This is a more elegant and systematic approach than simple thresholding and is a clear strength from a methodological standpoint.

Weaknesses:
1.	Quality & Reproducibility: The quality of the experimental evaluation is severely compromised by major, unexplained discrepancies in the baseline results.
     o	The paper reports that its primary baseline, aTLAS [1], achieves 77.60% average accuracy on ViT-B/32. However, the original aTLAS paper reports 84.98% for the identical setup.
     o	Even more concerning is the 10+ percentage point drop in the zero-shot performance of the base model itself (37.99% in this paper vs. 48.14% in the original aTLAS paper).
     o	These significant and unaddressed discrepancies cast serious doubt on the validity of the entire experimental framework and render the paper's claims of outperformance unconvincing.
2.	Significance & Scope of Evaluation: The paper's contribution is significantly diminished by its insufficient experimental scope and its failure to compare against a sufficient set of relevant, contemporary baselines.
     o	Limited Task Diversity: The evaluation is confined to a single task type: task addition for image classification. To demonstrate the general applicability of a model merging framework, it is essential to evaluate it on a broader range of tasks. For example, experiments on task negation, a standard benchmark in task arithmetic, are missing. Furthermore, the evaluation is limited to the vision domain; demonstrating performance on NLP tasks, even on smaller models like RoBERTa, would greatly strengthen the paper's claims of generality.
     o	Lack of State-of-the-Art Baselines: The comparison is almost exclusively limited to a single method (aTLAS). methods like AdaMerging [2], and data-free approaches like WUDI-merging[3] are completely ignored. This lack of a comprehensive comparison makes it impossible to assess the actual significance and standing of VTvC in the current landscape.
3.	Quality & Clarity on Practicality: The paper avoids a critical discussion of its own practical limitations.
     o	The claim of having "no additional inference costs" is misleading, as this is standard for most task-vector merging methods.
     o	The paper fails to acknowledge or quantify the prohibitive merging cost of VTvC, which requires a forward pass through an inference network for every sample. This makes the method far more computationally expensive than its peers.
     o	Due to this high cost, the method is not scalable to large language models (LLMs). This severe limitation is not addressed, raising questions about the practical utility and significance of the proposed framework.
[1] Zhang, Frederic Z., et al. "Knowledge composition using task vectors with learned anisotropic scaling." Advances in Neural Information Processing Systems 37 (2024): 67319-67354.
[2] Yang E, Wang Z, Shen L, et al. Adamerging: Adaptive model merging for multi-task learning[J]. arXiv preprint arXiv:2310.02575, 2023.
[3] Cheng, Runxi, et al. "Whoever started the interference should end it: Guiding data-free model merging via task vectors." arXiv preprint arXiv:2503.08099 (2025).

---

> ### Author Rebuttal · Authors · 2025-07-31
>
> Thank you for your thoughtful feedback and for acknowledging the sound theoretical formulation involving our variational framework and Spike‑and‑Slab prior. We address your questions as follows.
>
> **Q1: Baseline Performance Discrepancy.**
>
> Thank you for mentioning the performance discrepancy, and we elaborate on the reason for you now. First of all, our experimental setup is fair as all models are compared under the same experimental conditions, including the same datasets, the same preprocessing procedures, and the same evaluation metrics.
>
> The performance discrepancy is mainly due to the data augmentation strategies we used in the original manuscript. As we stated in  our paper (lines 264-266):
>
> > "To improve computational efficiency, we precompute feature representations for all datasets and apply data augmentation to generate multiple feature versions, thereby constructing a more diverse training set."
>
> This augmentation strategy is static and relatively weak compared to the dynamic data augmentation strategy used by the original aTLAS method (applying random transformations to each sample in real-time during training) and therefore caused accuracy decline. We adopted the static one purely for the sake of computational efficiency, and it presumably affects the performance of all models equally.
> To demonstrate this, in this response, we conduct experiments using the same dynamic data augmentation method as the original aTLAS paper. The results are shown in the following table:
>
> |Method|ViT-B/32|ViT-B/16|ViT-L/14|
> |-|:-:|:-:|:-:|
> |Zero-shot|48.22|55.33|64.72|
> |aTLAS|84.70|85.99|91.54|
> |VTVC **(Ours)**|**87.45**|**90.97** |**93.89**|
>
> Results show that after restoring the original augmentation strategy, the ViT-B/32, ViT-B/16, and ViT-L/14 models improved by **2.75%**, **4.98%**, and **2.35%** respectively compared to aTLAS. This proves that, regardless of experimental settings, our method consistently performs better.
> We will supplement the complete experimental results under identical data augmentation conditions in the revised version of our paper. We hope this addresses your concerns about the performance discrepancy.
>
> **Q2: Comprehensive Comparison and Trade-off Analysis.**
>
> We follow your suggestion and have conducted additional experiments with related methods for a comprehensive comparison. The results are reported as follows:
>
> |Methods|Cars|DTD|EuroSAT|GTSRB|MNIST|RESISC45|SUN397|SVHN|Avg.|
> |-|:-:|:-:|:-:|:-:|:-:|:-:|:-:|:-:|:-:|
> |Zero-shot|59.63|44.13|45.74|32.60|48.26|60.27|63.53|31.63|48.22|
> |Task Arithmetic(ICLR’23)|60.23|55.18|80.26|66.21|95.94|69.17|64.09|76.27|70.92|
> |AdaMerging(ICLR’24)|56.90|50.10|83.40|82.40|95.70|73.10|60.80|87.30|73.71|
> |WUDI-Merging(ICML’25)|70.88|71.53|**95.74**|**93.12**|98.01|86.21|**72.35**|**92.01**|84.98|
> |VTVC **(Ours)**|**76.02**|**86.88**|95.48|90.57|**98.24**|**91.35**|71.19|89.53|**87.45**|
>
> **(a) Final accuracy**
>
> On 8 visual benchmarks, VTVC achieves **state-of-the-art performance** with an average accuracy of 87.45%.
>
> **(b) Merging cost (time/compute)**
>
> The training and inference costs of the four methods (based on 1000 samples, running on a single NVIDIA A40 with a ViT-B/32 model and batch size 64) are shown in the table below:
>
> |Method|Training cost|Inference cost|
> |-|:-:|:-:|
> |Task Arithmetic(ICLR'23)|~0s|120s|
> |AdaMerging(ICLR'24)|~10s|5.6s|
> |WUDI-Merging(ICML'25)|~1s|5.6s|
> |VTVC **(Ours)**|112s|5.6s|
>
> Due to VTVC's structural design, the training stage requires learning a small number of additional parameters ${\Phi, \Psi}$. However, during inference, there is no need to traverse composition coefficients, significantly improving inference efficiency. Overall, VTVC trades a slightly higher one-time offline cost for significantly higher accuracy and sample adaptation capability, while maintaining millisecond-level online latency.
>
> **(c) Scalability**
>
> The scalability discussion for the four methods is shown in the table below:
>
> + **Larger Models**
>     + *Task Arithmetic*: Inference cost increases linearly with number of layers, grid search becomes even slower.
>     + *AdaMerging*: Number of parameter blocks increases, linear solver matrix becomes significantly larger.
>     + *WUDI-merging*: Computation scales linearly with parameters.
>     + *VTVC(Ours)*: No additional parameters added, inference latency still <1ms.
> + **More Tasks**
>     + *Task Arithmetic*: Requires complete grid search for $\lambda$ for new tasks.
>     + *AdaMerging*: Must re-solve all block-level coefficients.
>     + *WUDI-merging*: Direct SVD merging on $\tau_{new}$.
>     + *VTVC(Ours)*: Only requires few iterations of fine-tuning $\Phi$ to quickly absorb $\tau_{new}$.
>
> The results in Figure 5 and Appendix C.1 show that our method VTVC achieves better gating effects and model merging performance when transferred to larger backbone networks with only minimal additional cost. For new tasks, there is no need to traverse composition coefficients again, maintaining fast inference time.
>
> **(d) Data dependency**
>
> + *WUDI-merging*: Completely data-free, most lightweight, but cannot optimize for specific samples.
> + *AdaMerging*: Requires small amounts of validation data to estimate block-level coefficients, low data threshold, but lacks sample-level adaptation.
> + *VTVC(Ours)*: Requires rich validation data to learn coefficient mapping, suitable for data-rich or continuous incremental learning scenarios.
> Therefore, if pursuing highest accuracy and cross-domain generalization, VTVC's data investment is reasonable and necessary. In extremely data-scarce scenarios, WUDI-Merging or AdaMerging can be chosen as cost-accuracy trade-offs.
> We will provide complete experimental configurations, cost calculations, and validation results in the revised version, and highlight the above trade-off analysis in the main text discussion.
>
> **Q3: Broadening the Experimental Scope.**
>
> We strongly agree with your viewpoint and have promptly conducted supplementary experiments on task negation and the NLP domain. The specific results are as follows:
>
> + **Task negation**
>
>     In the task negation setting, the model aims to forget the target task as much as possible while maintaining control tasks within a certain accuracy range (>95%).
>
>     For *task negation*, we introduce an asymmetric gated sampling mechanism. This mechanism sets lower thresholds for target task coefficients while protecting important components from being filtered out. The experimental results for task negation using ViT-B/32 are as follows:
>
>     |Methods|Target($\downarrow$)|Control($\uparrow$)|
>     |-|:-:|:-:|
>     | Task Arithmetic(ICLR’23)|22.62|60.73|
>     | TIES-Merging(NeurIPS’23)|20.93|**61.52**|
>     | TaLoS(ICLR’25)|17.28|61.33|
>     | aTLAS(NeurIPS’24)|19.42|61.31|
>     | VTVC **(Ours)**|**16.75**|60.84|
>
>     Our experiments show that the asymmetric gating mechanism achieves better forgetting effects on target tasks while keeping control tasks within the required performance range. This demonstrates that our method works well not only for task addition but also for task negation.
>
> + **NLP domain**
>
>     For *discriminative language tasks*, we use RoBERTa as the backbone and evaluate on the 8-task GLUE benchmark. We will provide detailed experimental settings and more comprehensive results in the revised version.
>
>     |Methods|COLA|STS-2|MRPC|STS-B|QQP|QNLI|MNLI|RTE|Avg.|
>     |-|:-:|:-:|:-:|:-:|:-:|:-:|:-:|:-:|:-:|
>     |Zero-shot|0.00|53.76|85.01|4.01|37.48|53.05|37.09|71.19|42.70|
>     |Weight Averaging|0.00|59.21|85.79|46.99|45.37|63.94|48.00|71.19|52.56|
>     |Task Arithmetic(ICLR'23)|8.35|88.26|**89.57**|32.84|82.03|85.40|75.54|80.43|67.80|
>     |TIES-Merging(NeurIPS'23)|31.76|88.86|86.18|10.94|61.05|**85.94**|**83.01**|69.56|64.66|
>     |DARE(ICML'24)|0.00|88.14|86.61|30.19|**84.33**|79.09|63.95|77.16|63.68|
>     |VTVC **(Ours)**|**80.60**|**90.60**|88.48|**74.37**|80.40|83.20|74.00|**83.80**|**81.93**|
>
>     The experimental results further demonstrate that VTVC is also applicable to discriminative NLP tasks, showing cross-modal generality and robustness.
>
> **Q4: Justification for Sample-Specific Complexity.**
>
> We acknowledge that sample-level composition introduces additional computational overhead, primarily during inference-time coefficient prediction. However, our model amortizes this cost via a lightweight inference network, which outputs coefficients in a single forward pass and does not require iterative optimization. The results in **[Q2]** show that compared to task-level methods commonly used in existing task arithmetic work, VTVC can complete coefficient learning with minimal time during the inference stage after training, without trailing composition coefficients.
> Meanwhile, as illustrated in Figure 2 of our paper, sample-level methods possess the ability to capture and identify differences between samples that task-level methods cannot achieve. Experimental results demonstrate that a more fine-grained use of sample features can significantly improve model merging performance. Based on this finding, we believe the following scenarios are more suitable for deploying sample-level methods:
> 1. **Online inference/edge deployment scenarios** with mixed input sources, where sample-level methods can capture distinctive features of new data and provide more accurate predictions.
> 2. **High-risk or continual learning scenarios** (such as medical imaging, autonomous driving) that require evaluating decision confidence, where our method can smoothly absorb new domain knowledge while maintaining performance on old tasks.
> Based on the above discussion, we believe the additional cost of sample-level methods is controllable in many real-world complex scenarios and brings significant accuracy and generalization benefits.
>
> **Thank you again for helping us improve the paper, and hope our response can resolve your concerns!**

---

> > ### Comment · Reviewer_TZ6C · 2025-08-07
> >
> > Thanks for addressing my concerns. I will raise the score accordingly.

---

> > > ### Author Response · Authors · 2025-08-08
> > >
> > > We are deeply grateful for your constructive feedback and the score increase. Your insights have significantly strengthened our work, and we will incorporate all the additional experiments and clarifications discussed in our revised manuscript.

---

### Decision · Program_Chairs · 2025-09-17

**Decision:**

Accept (poster)

**Comment:**

This paper proposes a variational Bayesian framework of task vector composition for model merging, where combination coefficients are considered as sample-specific variables. The proposed method uses a spike-and-slab prior for sparsity, by which interpretability and generalization ability is improved. The work is technically solid. The experimental results demonstrate the effectiveness of the proposed method. Applying variational inference to the task vector composition problem is interesting. The scalability of the proposed method is a weakness. By the author response, most of the reviewers’ concerns were addressed. The additional experiments strengthen the paper.